# Statistical Inference under Performativity

**Xiang Li**[*]
Independent Researcher
xiangli0233@gmail.com

**Yunai Li**[*]
Northwestern University
yunaili2030@u.northwestern.edu

**Huiying Zhong**[*]
MIT
zhong826@mit.edu

**Lihua Lei**
Stanford University
lihualei@stanford.edu

**Zhun Deng**
UNC at Chapel Hill
zhundeng@cs.unc.edu

## Abstract

Performativity of predictions refers to the phenomenon where prediction-informed decisions influence the very targets they aim to predict—a dynamic commonly observed in policy-making, social sciences, and economics. In this paper, we *initiate* an end-to-end framework of statistical inference under performativity. Our contributions are twofold. First, we establish a central limit theorem for estimation and inference in the performative setting, enabling standard inferential tasks such as constructing confidence intervals and conducting hypothesis tests in policy-making contexts. Second, we leverage this central limit theorem to study prediction-powered inference (PPI) under performativity. This approach yields more precise estimates and tighter confidence regions for the model parameters (i.e., policies) of interest in performative prediction. We validate the effectiveness of our framework through numerical experiments. To the best of our knowledge, this is the first work to establish a complete statistical inference under performativity, introducing new challenges and inference settings that we believe will provide substantial value to policy-making, statistics, and machine learning.

## 1 Introduction

Prediction-informed decisions are ubiquitous in nearly all areas and play important roles in our daily lives. An important and commonly observed phenomenon is that prediction-informed decisions can impact the targets they aim to predict, which is called *performativity* of predictions. For instance, policies about loans based on default risk prediction can alter consumption habits of the population that will further have an impact on their ability to pay off their loans.

To characterize performativity of predictions, a rich line of work on performative prediction [25] have been formalizing and investigating this idea that predictive models used to support decisions can impact the data-generating process. Mathematically, given a parameterized loss function $\ell$, the aim of performative prediction is to optimize the performative risk:

$$\mathrm{PR}(\theta) := \mathbb{E}_{z \sim \mathcal{D}(\theta)} \ell(z; \theta) \tag{1}$$

where $z = (x, y) \in \mathcal{X} \times \mathcal{Y}$ is the input and output pair drawn from a distribution $\mathcal{D}(\theta)$ that is dependent on the loss parameter $\theta$. Typically, $\mathcal{D}(\theta)$ is unknown and the optimization objective $\mathrm{PR}(\theta)$ can be non-convex even if $\ell(z; \theta)$ is convex in $\theta$. Thus, finding a *performative optimal* point $\theta_{\mathrm{PO}} \in \arg\min_\theta \mathrm{PR}(\theta)$ (there might exist multiple optimizers due to non-convexity) can be theoretically intractable unless we impose very strong distributional assumptions [20]. As an

---

[*] for equal contribution, listed in alphabetical order.

39th Conference on Neural Information Processing Systems (NeurIPS 2025).

alternative, [25] mainly study how to obtain a *performative stable* point, which satisfies the following relationship:

$$\theta_{\text{PS}} = \arg\min_\theta \mathbb{E}_{z \sim \mathcal{D}(\theta_{\text{PS}})} \ell(z; \theta).$$

The performative stable point could be proved unique under some regularity conditions, and it could be shown close to a performative optimal point when the distribution shift between different $\theta$'s is not too dramatic, which makes it a good proxy to $\theta_{\text{PO}}$. In particular, $\theta_{\text{PS}}$ could be considered as a good proxy to the Stackelberg equilibrium in the strategic classification setting. Moreover, it could be calculated in distribution-agnostic settings.

Previous work mainly focuses on prediction performance and convergence rate analysis for performative prediction. On the contrary, another important aspect, ***inference under performativity***, eludes the literature. Although a central limit theorem was established in [7] for the stochastic gradient update algorithm with a single sample under performativity (a setting that is often impractical in applications such as policy-making), the authors assumed all structural knowledge to be given and did not provide any data-driven methods for estimating the covariance of the various quantities appearing in their central limit theorem. Therefore, [7] ***did not provide a complete statistical inference framework under performativity***.

However, inference is extremely important in performative prediction because parameter $\theta$ in many scenarios represents a concrete policy, such as a tax rate or credit score cutoff. Thus, when it comes to policy-making, the aim of tackling $\text{PR}(\theta)$ is not just for prediction, but more for obtaining a good policy. As a result, knowing convergence to $\theta_{\text{PS}}$ is not enough, we need to build statistical inference for $\theta_{\text{PS}}$ so as to enable people to report additional critical information like confidence or conduct necessary hypothesis testing.

**Our contributions.** In light of the importance of building statistical inference under performativity, in this work, we build a framework including the following elements.

(1). As our first contribution, we investigate a widely applied iterative algorithm to calculate $\theta_{\text{PS}}$, i.e., repeated risk minimization (RRM) (see details in Section 2), and establish central limit theorems for the $\widehat{\theta}_t$'s obtained in the RRM process towards $\theta_{\text{PS}}$. Based on that, we are able to obtain the confidence region for $\theta_{\text{PS}}$. Our results could be viewed as generalizing standard statistical inference from a ***static*** setting to a ***dynamic*** setting.

(2). As our second contribution, we further leverage the derived central limit theorems to investigate prediction-powered inference (PPI), another recently popular topic in modern statistical learning, under performativity. Our results generalize previous work [1] to a dynamic performative setting. This enables us to obtain better estimation and inference for the RRM process and $\theta_{\text{PS}}$. More importantly, our results could also help mitigate ***data scarcity*** issues in getting feedback about policy implementation that often conducted by doing surveys that frequently encounter non-responses [13]. Thus, we also contribute to generalizing the line of work on performative prediction by introducing a ***more data efficient*** algorithm.

To sum up, our work establishes the first end-to-end framework for statistical inference under performativity for the celebrated repeated risk minimization algorithm. Meanwhile, we introduce prediction-powered inference under performativity to enable a more efficient inference. We believe our work would inspire new interesting topics and bring up new challenges to both areas of perforamative prediction and prediction-powered inference, as well as add significant value to policy-making in a broad range of areas such as social science and economics.

## 1.1 Related Work

**Performative prediction.** Performativity describes the phenomenon whereby predictions influence the outcomes they aim to predict. [25] were the first to formalize performative prediction in the supervised-learning setting; their work, along with the majority of subsequent papers [5, 9, 16, 19, 21, 24, 26, 29], have been focused on performative stability and proposed algorithms for learning performative stable parameters. On the other hand, performative optimality requires much stricter conditions (e.g. distributional assumptions to ensure the convexity of $\text{PR}(\theta)$) than performative stability, a few papers address the problem of finding performative optimal parameters. [20] introduce a two-stage method that learns a distribution map to locate the performative optimal parameter. [17] study performative optimality in outcome-only performative settings. Finally, [11]

provide a comprehensive overview of learning algorithms, optimization methods, and applications for performative prediction. Unlike these prior works on performative prediction, which focus on prediction accuracy, our work is dedicated to constructing powerful and statistically valid inference procedures under the performative framework.

**Prediction-powered inference.** [2] first introduced the prediction-powered inference (PPI) framework, which leverages black-box machine learning models to construct valid confidence intervals (CIs) for statistical quantities. Since then, PPI has been extended and applied in various settings. Closely related to our strategies, [1] propose PPI++, a more computationally efficient procedure that enhances predictability by accommodating a wider range of models on unlabeled data, while guaranteeing performance (e.g. CI width) no worse than that of classical inference methods. Other extensions include Stratified PPI [10], which incorporates simple data stratification strategies into basic PPI estimates; Cross PPI [33], which obtains confidence intervals with significantly lower variability by including model training; Bayesian PPI [12] and FAB-PPI [6], which propose frameworks for PPI based on Bayesian inference. PPI is also connected to topics such as semi-parametric inference and missing-data imputation [8, 27, 28, 30]. Our work is the first one to study PPI under performativity, and we validate the PPI framework in the performative setting both theoretically and empirically.

**Inference in performativity.** [5] studies identifiability and estimation error under a specific micro-foundation model with performativity. Yet it doesn't address confidence interval construction. A closely related work is [7], which also establishes the asymptotic normality and minimax optimality for performative settings. It focuses on stochastic gradient update with one sample per iteration, whereas our work analyzes the empirical risk minimizer on batch updates. Moreover, [7] does not provide a data-driven approach for covariance estimation and thus lacks an end-to-end inference framework. In contrast, our work explicitly handles density estimation and provides an end-to-end inference method for constructing confidence intervals, which is missing in existing literature, to the best of our knowledge.

## 2    Background

In this section, we recap more detailed background knowledge about performative prediction and prediction-powered inference.

**Repeated risk minimization.** Recall that the main objective of interest is the *performative stable* point, which satisfies the following relationship:

$$\theta_{\mathrm{PS}} = \arg\min_{\theta} \mathbb{E}_{z \sim \mathcal{D}(\theta_{\mathrm{PS}})} \ell(z; \theta).$$

Repeated risk minimization (RRM) is a simple algorithm that can efficiently find $\theta_{\mathrm{PS}}$. Specifically, one starts with an arbitrary $\theta_0$ and repeat the following procedure:

$$\theta_{t+1} = \arg\min_{\theta} \mathbb{E}_{z \sim \mathcal{D}(\theta_t)} \ell(z; \theta)$$

for $t \in \mathbb{N}$. Under some regularity conditions, the above update is well-defined and provably converges to a unique $\theta_{\mathrm{PS}}$ at a linear rate.

**Theorem 2.1** (Informal, adopted from [25]). *If the loss is smooth, strongly convex, and the mapping $\mathcal{D}(\cdot)$ satisfies certain Lipchitz conditions, then $\theta_{PS}$ is uniquely defined and repeated risk minimization converges to $\theta_{PS}$ in a linear rate.*

We will further explicitly state those conditions in Section 3. Throughout the paper, we will mainly focus on building an inference framework under the repeated risk minimization algorithm.

**Prediction-powered inference.** A rich line of work on prediction-powered inference (PPI) [1] considers how to combine limited gold-standard labeled data with abundant unlabeled data to obtain more efficient estimation and construct tighter confidence regions for some unknown parameters. Specifically, a general predictive setting is considered in which each instance has an input $x \in \mathcal{X}$ and an associated observation $y \in \mathcal{Y}$. People have access to a limited set of gold-standard labeled data $\{x_i, y_i\}_{i=1}^n$ that are i.i.d. drawn from a distribution $\mathcal{D}$. Meanwhile, we have abundant unlabeled data $\{x_i^u\}_{i=1}^N$ that are i.i.d. drawn from the same marginal distribution as gold-standard labeled data, i.e. $\mathcal{D}_{\mathcal{X}}$, where $N \gg n$. In addition, an annotating model $f : \mathcal{X} \mapsto \mathcal{Y}$ (possibly off-the-shelf and black-

box machine learning models) is used to label data[1]. In [1], the authors show that for a convex loss with a unique solution, compared with standard M-estimator $\widehat{\theta}^{\text{SL}} = \arg\min_\theta \sum_{i=1}^n \ell(x_i, y_i; \theta)/n$,

$$\widehat{\theta}^{\text{PPI}}(\lambda) := \arg\min_\theta \lambda \frac{1}{N} \sum_{i=1}^N \ell(x_i^u, f(x_i^u); \theta) + \frac{1}{n} \sum_{i=1}^n \ell(x_i, y_i; \theta) - \lambda \frac{1}{n} \sum_{i=1}^n \ell(x_i, f(x_i); \theta)$$

can be a better estimator of $\theta^* = \arg\min_\theta \mathbb{E}_{z\sim\mathcal{D}}\ell(z; \theta)$ via appropriately chosen $\lambda$ based on data.

**Notation.** For $K \in \mathbb{N}_+$, we use $[K]$ to denote $\{1, 2, \cdots, K\}$. We use $\xrightarrow{P}$ and $\xrightarrow{D}$ to denote convergence in probability and in distribution, respectively. For two set $\mathcal{S}$ and $\mathcal{S}'$, we use $\mathcal{S} + \mathcal{S}'$ to denote the set $\{s + s' : s \in \mathcal{S}, s' \in \mathcal{S}'\}$. $\mathcal{N}(\mu, \Sigma)$ denotes a Gaussian distribution with mean $\mu$ and covariance matrix $\Sigma$. Lastly, we denote a $k$-dimensional identity matrix as $I_k$. We use $\mathcal{B}(c, r)$ to denote a ball with center $c$ and radius $r$. Lastly, we use $\|\cdot\|$ for $\ell_2$-norm and $\mathbf{1}$ to denote a column vector with all coordinates 1.

## 3  Inference under Performativity

In this section, we initiate the inference framework for repeated risk minimization in the batch setting under performativity. We mainly consider the repeated risk minimization setting. Unlike standard inference problems, where estimators are built for a ***fixed*** underlying data distribution, in our ***dynamic*** setting specified below, building asymptotic results such as CLT imposes extra challenges and this has not been covered by any existing literature so far.

Specifically, at time $t = 0$, we have access to a set of labeled data $\{z_{0,i}\}_{i=1}^{n_0}$ that is i.i.d. drawn from a distribution $\mathcal{D}(\theta_0)$, where $z_{0,i} = (x_{0,i}, y_{0,i})$ and $\theta_0$ is chosen by us. Then, we use the empirical repeated risk minimization to output

$$\widehat{\theta}_1 = \arg\min_\theta \frac{1}{n_0} \sum_{i=1}^{n_0} \ell(z_{0,i}; \theta)$$

as an estimator of $\theta_1 = \arg\min_\theta \mathbb{E}_{z\sim\mathcal{D}(\theta_0)}\ell(z; \theta)$. Then, for $t \geqslant 1$, at time $t$, we further have access to a set of labeled data $\{z_{t,i}\}_{i=1}^{n_t}$ that are i.i.d. drawn from the distribution induced by last iteration, i.e., $\mathcal{D}(\widehat{\theta}_t)$. Let us further define $G(\tilde{\theta}) = \arg\min_\theta \mathbb{E}_{z\sim\mathcal{D}(\tilde{\theta})}\ell(z; \theta)$. Then, we can obtain the output

$$\widehat{\theta}_{t+1} = \arg\min_\theta \frac{1}{n_t} \sum_{i=1}^{n_t} \ell(z_{t,i}; \theta)$$

as an estimator of $\theta_{t+1} = G(\theta_t)$. This iterative process will incur two trajectories, i.e., (1) ***underlying trajectory***: $\theta_0 \to \theta_1 \to \cdots \theta_t \to \cdots$; (2) ***trajectory in practice***: $\theta_0 \to \widehat{\theta}_1 \to \cdots \widehat{\theta}_t \to \cdots$.

Our aim is to provide inference on $\widehat{\theta}_t$ for any $t \geqslant 1$. For simplicity, we let $n_t = n$ for all $t$.

### 3.1  Central Limit Theorem of $\widehat{\theta}_t$

In order to build CLT for $\widehat{\theta}_t$, we first establish the consistency of $\widehat{\theta}_t$, which is relatively straightforward given that [25] has built the non-asymptotic convergence results. Then, we introduce our main result on building CLT for $\widehat{\theta}_t$. Lastly, we provide a novel method to estimate the variance of $\widehat{\theta}_t$.

**Consistency of $\widehat{\theta}_t$.** We start with proving the consistency of $\widehat{\theta}_t$. Recall that we have a trajectory induced by the samples $\theta_0 \to \widehat{\theta}_1 \to \cdots \widehat{\theta}_t \to \cdots$ by the iterative algorithm deployed. Without consistency, CLT is not expected to hold. Our results are based on the following assumptions.

**Assumption 3.1.** Assume the loss function $\ell$ satisfies:

(a). (*Local Lipschitzness*) Loss function $\ell(z; \theta)$ is locally Lipschitz: for each $\theta$, there exist a neighborhood $\Upsilon(\theta)$ of $\theta$ such that $\ell(z; \tilde{\theta})$ is $L(z)$ Lipschitz w.r.t $\tilde{\theta}$ for all $\tilde{\theta} \in \Upsilon(\theta)$ and $\mathbb{E}_{z\sim\mathcal{D}(\theta)}L(z) < \infty$.

---

[1] The annotating function $f$ could either be a stochastic or deterministic function. It could even take other inputs besides $x$, but for simplicity, we only consider the annotation with the form $f(x)$.

(b). (*Joint Smoothness*) Loss function $\ell(z;\theta)$ is $\beta$-jointly smooth in both $z$ and $\theta$:

$$\|\nabla_\theta \ell(z;\theta) - \nabla_\theta \ell(z;\theta')\|_2 \leqslant \beta \|\theta - \theta'\|_2\,, \quad \|\nabla_\theta \ell(z;\theta) - \nabla_\theta \ell(z';\theta)\|_2 \leqslant \beta \|z - z'\|_2\,,$$

for any $z, z' \in \mathcal{Z}$ and $\theta, \theta' \in \Theta$.

(c). (*Strong Convexity*) Loss function $\ell(z;\theta)$ is $\gamma$-strongly convex w.r.t $\theta$:

$$\ell(z;\theta) \geqslant \ell(z;\theta') + \nabla_\theta \ell(z;\theta')^\top (\theta - \theta') + \frac{\gamma}{2}\|\theta - \theta'\|^2\,,$$

for any $z \in \mathcal{Z}$ and $\theta, \theta'$.

(d). ($\varepsilon$-*Sensitivity*) The distribution map $D(\theta)$ is $\varepsilon$-sensitive, i.e.:

$$W_1\left(\mathcal{D}(\theta), \mathcal{D}(\theta')\right) \leqslant \varepsilon \|\theta - \theta'\|,$$

for any $\theta, \theta'$, where $W_1$ is the Wasserstein-1 distance.

**Remark 3.2.** *The assumptions (b), (c), (d) follow the standard ones in [25], which are proved to be the minimal requirements for trajectory convergence. We additionally require (a) to build consistency for $\widehat{\theta}_t$ beyond convergence of $\widehat{\theta}_t$ to $\theta_{PS}$.*

**Proposition 3.3.** *Under Assumption 3.1, if $\varepsilon < \frac{\gamma}{\beta}$, then for any given $T \geqslant 0$, we have that for all $t \in [T]$,*

$$\widehat{\theta}_t \xrightarrow{P} \theta_t.$$

**Building CLT for $\widehat{\theta}_t$.** In order to build the central limit theorem for $\widehat{\theta}_t$, we need to introduce a few extra assumptions. Due to limited space, going forward, we defer the required assumptions in later theorems to Appendix.

Let us denote $\Sigma_{\tilde{\theta}}(\theta) = H_{\tilde{\theta}}(\theta)^{-1} V_{\tilde{\theta}}(\theta) H_{\tilde{\theta}}(\theta)^{-1}$, where $H_{\tilde{\theta}}(\theta) = \nabla_\theta^2 \mathbb{E}_{z \sim \mathcal{D}(\tilde{\theta})} \ell(z;\theta)$ and $V_{\tilde{\theta}}(\theta) = \mathrm{Cov}_{z \sim \mathcal{D}(\tilde{\theta})}\left(\nabla_\theta \ell(z;\theta)\right)$. And recall that $G(\tilde{\theta}) = \arg\min_\theta \mathbb{E}_{z \sim \mathcal{D}(\tilde{\theta})} \ell(z;\theta)$.

**Theorem 3.4** (Central Limit Theorem of $\widehat{\theta}_t$). *Under Assumption 3.1 and A.1, if $\varepsilon < \frac{\gamma}{\beta}$, then for any given $T \geqslant 0$, we have that for all $t \in [T]$,*

$$\sqrt{n}(\widehat{\theta}_t - \theta_t) \xrightarrow{D} \mathcal{N}(0, V_t)$$

*with*

$$V_t = \sum_{i=1}^t \left[\prod_{k=i}^{t-1} \nabla G(\theta_k)\right] \Sigma_{\theta_{i-1}}(\theta_i) \left[\prod_{k=i}^{t-1} \nabla G(\theta_k)\right]^\top.$$

*In particular, $\nabla G(\theta_k) = -H_{\theta_k}(\theta_{k+1})^{-1}\left(\nabla_{\tilde{\theta}} \mathbb{E}_{z \sim \mathcal{D}(\theta_k)} \nabla_\theta \ell(z;\theta_{k+1})\right)$, where $\nabla_{\tilde{\theta}}$ is taking gradient for the parameter in $\mathcal{D}(\tilde{\theta})$, $\nabla_\theta$ is taking gradient for the parameter in $\ell(z;\theta)$ and $\prod_{k=t}^{t-1} \nabla G(\theta_k) = I_d$.*

**Estimation of $\nabla G(\theta_t)$, $V_t$** Given the established CLT for $\widehat{\theta}_t$, in order to construct confidence regions for $\widehat{\theta}_t$ in practice, the only thing left is to provide an estimation of $\nabla G(\theta_t)$ and $V_t$ with samples. In previous results, with a more detailed calculation, we obtain

$$\nabla G(\theta_k) = -H_{\theta_k}(\theta_{k+1})^{-1} \mathbb{E}_{z \sim \mathcal{D}(\theta_k)}[\nabla_\theta \ell(z, \theta_{k+1}) \nabla_\theta \log p(z, \theta_k)^\top].$$

where $p(\cdot, \theta)$ is the density function of distribution $D(\theta)$, and the score function $\nabla_\theta \log p(z, \theta)$ is thus a $d$-dimensional vector (recall that $\theta$ is of dimension $d$). In order to estimate it for any $\theta$, we propose a novel score matching method. Specifically, we use a model $M(z, \theta; \psi)$ parameterized by $\psi$ to approximate $p(z, \theta)$. Inspired by the objective in [14], for any given $\theta$ (e.g., $\widehat{\theta}_t$), we aim to minimize the following objective parameterized by $\psi$ **for all** $\theta$:

$$J(\theta; \psi) = \int p(z, \theta) \|\nabla_\theta \log p(z, \theta) - s(z, \theta; \psi)\|^2 dz$$

$$= \int p(z, \theta)\left(\|\nabla_\theta \log p(z, \theta)\|^2 + \|s(z, \theta; \psi)\|^2 - 2\nabla_\theta \log p(z, \theta)^\top s(z, \theta; \psi)\right)dz$$

where $s(z, \theta; \psi) = \nabla_\theta \log M(z, \theta; \psi)$. If we can learn a $\widehat{\psi}$ so that $s(z, \theta; \widehat{\psi}) = \nabla_\theta \log p(z, \theta)$ for all $\theta$, then we can reach the minimum $J(\theta; \widehat{\psi}) = 0$.

Notice the first term is unrelated to $\psi$; the second term involves the model $M$ that is chosen by us, so we have the analytical expression of it. Thus, our key task will be estimating the third term, which involves $\mathcal{K}(\theta; \psi) := \int p(z, \theta) \nabla_\theta \log p(z, \theta)^\top s(z, \theta; \psi) dz$.

We remark that in our setting, instead of taking the gradient at $z$, we have new challenges in taking the gradient at $\theta$. So, we derive the following key lemma.

**Lemma 3.5.** *Under Assumption A.2, we have*

$$\mathcal{K}(\theta; \psi) = \sum_{i=1}^{d} \left[ \frac{\partial}{\partial \theta^{(i)}} \int p(z, \theta) \frac{\partial \log M(z, \theta; \psi)}{\partial \theta^{(i)}} dz - \int p(z, \theta) \frac{\partial^2 \log M(z, \theta; \psi)}{\partial \theta^{(i)2}} dz \right]$$

*where $\theta^{(i)}$ is the $i$-th coordinate of $\theta$.*

Based on the lemma, we propose a novel gradient-free score matching method with *policy perturbation* to estimate $\mathcal{K}(\widehat{\theta}_t; \psi)$ for any $t \in [T]$. Policy perturbation is a commonly used technique in estimating the policy effect under general equilibrium shift [22] or interference [32]. Instead of just getting samples for $\widehat{\theta}_t$ for each $t$, we additionally sample for all perturbed policies in $\{\widehat{\theta}_t + \eta e_1, \widehat{\theta}_t + \eta e_2, \cdots, \widehat{\theta}_t + \eta e_d\}$, where $\eta > 0$ is a small scalar at our choice and $\{e_j\}_j$ are standard basis for $\mathbb{R}^d$. Typically, for a policy $\theta$, its dimension $d$ is low. One concrete example in practice is to use slightly different price strategies in different local markets.

Specifically, the term $\int p(z, \widehat{\theta}_t) \frac{\partial^2 \log M(z, \widehat{\theta}_t; \psi)}{\partial \theta^{(i)2}} dz$ could be easily estimated by using empirical mean, e.g., $\frac{1}{n} \sum_{j=1}^{n} \frac{\partial^2 \log M(z_{t,j}, \widehat{\theta}_t; \psi)}{\partial \theta^{(i)2}}$. And for the derivative $\frac{\partial}{\partial \theta^{(i)}} \int p(z_{t,j}, \widehat{\theta}_t) \frac{\partial \log M(z, \widehat{\theta}_t; \psi)}{\partial \theta^{(i)}} dz$, if we draw additional $k$ samples $\{z_{t,u}^{(i)}\}_{u=1}^{k}$ for each perturbed policy $\widehat{\theta}_t + \eta e_i$, we can use the following estimator:

$$\frac{1}{\eta} \left( \frac{1}{k} \sum_{u=1}^{k} \frac{\partial \log M(z_{t,u}^{(i)}, \widehat{\theta}_t + \eta e_i; \psi)}{\partial \theta^{(i)}} - \frac{1}{n} \sum_{u=1}^{n} \frac{\partial \log M(z_{t,u}, \widehat{\theta}_t; \psi)}{\partial \theta^{(i)}} \right).$$

Combining the above, we have a straightforward way to estimate $G(\theta_t)$ and $V_t$ for any $t \in [T]$ by plugging in the empirical estimate. Let us denote the estimator of $V_t$ by $\widehat{V}_t$. Then, we would have

$$\widehat{V}_t^{-1/2} \sqrt{n}(\widehat{\theta}_t - \theta_t) \xrightarrow{D} \mathcal{N}(0, I_d) \tag{2}$$

if the model $M(z, \theta; \psi)$ is expressive enough.

**Theoretical validity of estimation.** Now we prove that the policy perturbation method provides a valid estimator of $\nabla G(\theta_k)$. Recall that the gradient of $G$ is given by

$$\nabla G(\theta_k) = -H_{\theta_k}(\theta_{k+1})^{-1} \mathbb{E}_{\theta_k}[\nabla_\theta \ell(z; \theta_{k+1}) \nabla_\theta \log p(z, \theta_k)^\top],$$

and the estimator is defined by

$$\widehat{g}_k := -H_{\widehat{\theta}_k}(\widehat{\theta}_{k+1})^{-1} \widehat{\mathbb{E}}_{\widehat{\theta}_k}[\nabla_\theta \ell(z; \widehat{\theta}_{k+1}) s(z, \widehat{\theta}_k; \widehat{\psi}(\widehat{\theta}_k))^\top].$$

**Theorem 3.6.** *Under Assumption 3.1 A.1, A.9 and A.11, we have*

$$\|\widehat{g}_k - \nabla G(\theta_k)\|^2 = O_p(\frac{1}{\sqrt{n}} + \frac{1}{\eta \sqrt{\min(n, k)}} + \eta + a_n).$$

*where $a_n = o(1)$ is a vanishing optimization error term defined in Assumption A.9(d).*

The proof is by decomposing the error between the empirical and true gradient into several components, where each varies in whether it uses empirical or population quantities. We bound the deviations between the true parameters $\theta_k$, $\theta_{k+1}$ and their empirical counterparts $\widehat{\theta}_k$, $\widehat{\theta}_{k+1}$ using Theorem 3.4. We bound the true score function and its estimation via a uniform bound of $\|\nabla \log p(z, \theta) - s(z, \theta, \psi(\theta))\|^2$ over the perturbed policies using Theorem A.13. Taken together, the above result shows the consistency of $\widehat{g}_k$ obtained by policy perturbation.

## 3.2 Bias-Aware Inference for Performative Stable Point

Finally, we further provide a way to construct the confidence region for $\theta_{\text{PS}}$. This is directly followed from our previous results on building CLT for $\widehat{\theta}_t$. By the convergence results derived for the underlying trajectory by [25], under Assumption 3.1, we have

$$\|\theta_t - \theta_{\text{PS}}\| \leqslant \left(\frac{\varepsilon\beta}{\gamma}\right)^t \|\theta_0 - \theta_{\text{PS}}\|.$$

Thus, we can immediately obtain the following corollary by using bias-aware inference – a commonly seen technique in econometrics [3, 4, 15, 23].

**Corollary 3.7** (Confidence region construction for $\theta_{\text{PS}}$). *Under Assumption 3.1, A.1, A.2, and A.3, if $\varepsilon < \frac{\gamma}{\beta}$, for any $\delta \in (0,1)$, we can obtain a confidence region $\widehat{\mathcal{R}}_t(n,\delta)$ for $\theta_t$ by using Eq. 2, such that*

$$\lim_{n\to\infty} \mathbb{P}\left(\theta_t \in \widehat{\mathcal{R}}_t(n,\delta)\right) = 1 - \delta.$$

*Moreover, if $\theta_0, \theta_{PS} \in \{\theta : \|\theta\| \leqslant B\}$,*

$$\lim_{n\to\infty} \mathbb{P}\left(\theta_{PS} \in \widehat{\mathcal{R}}_t(n,\delta) + \mathcal{B}\left(0, 2B\left(\frac{\varepsilon\beta}{\gamma}\right)^t\right)\right) \geqslant 1 - \delta.$$

Corollary 3.7 provides a way to construct the confidence region for the performative stable point based on the confidence region for $\theta_t$. Notice that the derived new confidence region is quite close to $\widehat{\mathcal{R}}_t(n,\delta)$ and the difference vanishes exponentially fast as $t$ grows. Thus, we expect the derived region to be quite tight for moderately large $t$, meaning:

$$\lim_{n\to\infty} \mathbb{P}\left(\theta_{\text{PS}} \in \widehat{\mathcal{R}}_t(n,\delta) + \mathcal{B}\left(0, 2B\left(\frac{\varepsilon\beta}{\gamma}\right)^t\right)\right) \approx 1 - \delta.$$

For the condition $\theta_0, \theta_{\text{PS}} \in \{\theta : \|\theta\| \leqslant B\}$, it will be natural to satisfy and we can get an explicit and feasible upper bound $B$ under mild conditions. It is because that $\theta_0$ is at our choice and we can further derive an explicit and feasible upper bound for $\|\theta_{\text{PS}}\|$ by using the strong convexity. Specifically, by $\gamma$-strong convexity of the loss function with respect to $\theta$, we have

$$\left(\mathbb{E}_{z\sim\mathcal{D}(\theta_{\text{PS}})}\nabla_\theta \ell(z;\theta_0) - \mathbb{E}_{z\sim\mathcal{D}(\theta_{\text{PS}})}\nabla_\theta \ell(z;\theta_{\text{PS}})\right)^\top (\theta_0 - \theta_{\text{PS}}) \geqslant \gamma \|\theta_0 - \theta_{\text{PS}}\|^2.$$

Since $\mathbb{E}_{z\sim\mathcal{D}(\theta_{\text{PS}})}\nabla_\theta \ell(z;\theta_{\text{PS}}) = 0$, this leads to $\|\mathbb{E}_{z\sim\mathcal{D}(\theta_{\text{PS}})}\nabla_\theta \ell(z;\theta_0)\| \geqslant \gamma \|\theta_0 - \theta_{\text{PS}}\|$. Thus, if we further have $\sup_{z\in\mathcal{Z}} \|\nabla_\theta \ell(z;\theta_0)\| \leqslant \tilde{B}$ for $\tilde{B} > 0$, which could be achieved and calculated by assuming $\mathcal{Z}$ is compact and use the continuity of $\nabla_\theta \ell(\cdot, \theta_0)$. Then, it will immediately give us an upper bound for $\|\theta_{\text{PS}}\|$ that $\|\theta_{\text{PS}}\| \leqslant \tilde{B}/\gamma + \|\theta_0\|$.

## 4 Prediction-Powered Inference under Performativity

In this section, we further investigate prediction-powered inference (PPI) under performativity to enhance estimation and obtain improved confidence regions for the model parameter (i.e., policy) under performativity. This can also address the data scarcity issue in human responses when doing a survey to get feedback on policy implementation.

Specifically, at time $t = 0$, besides the limited set of gold-standard labeled data $\{x_{0,i}, y_{0,i}\}_{i=1}^{n_0}$ that are i.i.d. drawn from a distribution $\mathcal{D}(\theta_0)$, we have abundant unlabeled data $\{x_{0,i}^u\}_{i=1}^{N_0}$ that are i.i.d. drawn from the same marginal distribution as gold-standard labeled data, i.e. $\mathcal{D}_{\mathcal{X}}(\theta_0)$, where $N_0 \gg n_0$. In addition, an annotating model $f : \mathcal{X} \mapsto \mathcal{Y}$ is used to label data [2], which leads to $\{x_{0,i}, f(x_{0,i})\}_{i=1}^{n_0}$ and $\{x_{0,i}^u, f(x_{0,i}^u)\}_{i=1}^{N_0}$. Then, we use the following mechanism to output

$$\widehat{\theta}_1^{\text{PPI}}(\lambda_1) = \arg\min_\theta \frac{\lambda_1}{N} \sum_{i=1}^N \ell(x_{0,i}^u, f(x_{0,i}^u); \theta) + \frac{1}{n} \sum_{i=1}^n \left(\ell(x_{0,i}, y_{0,i}; \theta) - \lambda_1 \ell(x_{0,i}, f(x_{0,i}); \theta)\right)$$

---

[2]Our theory can easily be extended to allow using different annotating function for each iteration, but for simplicity in presentation, we use $f$ for all iterations.

for a scalar $\lambda_1$ as an estimator of $\theta_1$. After that, for $t \geqslant 1$, at time $t$, besides having access to the set of gold-standard labeled data $\{x_{t,i}, y_{t,i}\}_{i=1}^{n_t}$ that are i.i.d. drawn from a distribution $\mathcal{D}(\widehat{\theta}_t)$. Meanwhile, we have abundant unlabeled data $\{x_{t,i}^u\}_{i=1}^{N_t}$ that are i.i.d. drawn from the same marginal distribution as gold-standard labeled data, i.e. $\mathcal{D}_{\mathcal{X}}(\widehat{\theta}_t)$, where $N_t \gg n_t$. Similar as before, we can estimate $\theta_{t+1}$ via

$$\widehat{\theta}_{t+1}^{\text{PPI}}(\lambda_{t+1}) = \arg\min_\theta \frac{\lambda_{t+1}}{N} \sum_{i=1}^N \ell(x_{t,i}^u, f(x_{t,i}^u); \theta) + \frac{1}{n} \sum_{i=1}^n \left( \ell(x_{t,i}, y_{t,i}; \theta) - \lambda_{t+1} \ell(x_{t,i}, f(x_{t,i}); \theta) \right)$$

for a scalar $\lambda_{t+1}$. This incurs a trajectory in practice: $\theta_0 \to \widehat{\theta}_1^{\text{PPI}}(\lambda_1) \to \cdots \widehat{\theta}_t^{\text{PPI}}(\lambda_t) \to \cdots$. Notice that if we choose $\lambda_t = 0$ for all $t \in [T]$, this will degenerate to the case in Section 3. Later on, we will demonstrate how to choose $\{\lambda_t\}_{t=1}^T$ via data to enhance inference. Our mechanism could be adaptive to the data quality with carefully chosen $\{\lambda_t\}_{t=1}^T$ and could be viewed as an extension of the classical PPI++ mechanism [25] to the setting under performativity.

**Building CLT for $\widehat{\theta}_t^{\text{PPI}}(\lambda_t)$.** We start with building the central limit theorem for $\widehat{\theta}_t^{\text{PPI}}(\lambda_t)$ with fixed constant scalars $\{\lambda_t\}_{t=1}^T$ for any $t \in [T]$. The proof is similar to that of Theorem 3.4. Specifically, we denote
$$\Sigma_{\lambda, \tilde{\theta}}(\theta; r) = H_{\tilde{\theta}}(\theta)^{-1} \left( r V_{\lambda, \tilde{\theta}}^f(\theta) + V_{\lambda, \tilde{\theta}}(\theta) \right) H_{\tilde{\theta}}(\theta)^{-1}$$

with $V_{\lambda, \tilde{\theta}}^f(\theta) = \lambda^2 \text{Cov}_{x \sim \mathcal{D}_{\mathcal{X}}(\tilde{\theta})}(\nabla_\theta \ell(x, f(x); \theta))$ and $V_{\lambda, \tilde{\theta}}(\theta) = \text{Cov}_{(x,y) \sim \mathcal{D}(\tilde{\theta})}(\nabla_\theta \ell(x, y; \theta) - \lambda \nabla_\theta \ell(x, f(x); \theta))$. Then, we have the following theorem.

**Theorem 4.1** (Central Limit Theorem of $\widehat{\theta}_t^{\text{PPI}}(\lambda_t)$). *Under Assumption 3.1, A.4, and A.5, if $\varepsilon < \frac{\gamma}{\beta}$ and $\frac{n}{N} \to r$ for some $r \geqslant 0$, then for any given $T \geqslant 0$, we have that for all $t \in [T]$,*
$$\sqrt{n}\big(\widehat{\theta}_t^{PPI}(\lambda_t) - \theta_t\big) \xrightarrow{D} \mathcal{N}\Big(0, V_t^{PPI}(\{\lambda_j, \theta_j\}_{j=1}^t; r)\Big)$$

*with*
$$V_t^{PPI}\big(\{\lambda_j, \theta_j\}_{j=1}^t; r\big) = \sum_{i=1}^t \left[\prod_{k=i}^{t-1} \nabla G(\theta_k)\right] \Sigma_{\lambda_i, \theta_{i-1}}(\theta_i; r) \left[\prod_{k=i}^{t-1} \nabla G(\theta_k)\right]^\top.$$

**Selection of parameters $\{\lambda_t\}_{t=1}^T$.** Now, the only thing left is to select $\{\lambda_t\}_{t=1}^T$, so as to enhance estimation and inference. As choosing $\lambda_t = 0$ for all $t \in [T]$ will degenerate to Theorem 3.4, we expect that we appropriately choose $\{\lambda_t\}_{t=1}^T$ to make $F(V_t) \geqslant F(V_t^{\text{PPI}}(\{\lambda_j, \theta_j\}_{j=1}^t; r))$, where $F$ is a user-specified scalarization operator depending on different aims. For instance, if we are interested in optimizing the sum of asymptotic variance of coordinates of $\theta_t$, then, $F(V_t) = \text{Tr}(V_t)$. Or if we are interested in the inference of the sum of all coordinates $\theta_t$, i.e., $\mathbf{1}^\top \theta_t$, then $F(V_t) = \mathbf{1}^\top V_t \mathbf{1}$.

We consider a greedy sequential selection mechanism as follows. Imagine that we have selected $\{\lambda_i^*\}_{i=1}^{t-1}$ via the data, and our aim is to select a $\lambda_t^*$ so as to make $F\left(V_t^{\text{PPI}}(\{\lambda_j^*, \theta_j\}_{j=1}^{t-1}, \lambda, \theta_t; r)\right) \geqslant F\left(V_t^{\text{PPI}}(\{\lambda_j^*, \theta_j\}_{j=1}^t; r)\right)$ for any $\lambda$. Thus, we choose
$$\lambda_t^* = \arg\min_\lambda F\left(V_t^{\text{PPI}}(\{\lambda_j^*, \theta_j\}_{j=1}^{t-1}, \lambda, \theta_t; r)\right). \tag{3}$$

However, in practice, there are still several issues that need to be addressed. First, we need to choose $\{\lambda_t\}_{t=1}^T$ via observations, and this could be handled by using our results in Section 3 to estimate $\nabla_\theta G(\theta)$, and we obtain a sample version $\widehat{\lambda}_t$ by plugging in the estimation. Second, when obtaining $\lambda_t^*$ in Eq. 3, we actually need $\theta_t$ in $\Sigma_{\lambda_i, \theta_{i-1}}(\theta_i; r)$. But when using samples to estimate, we need to get $\widehat{\lambda}_t$ first before we obtain $\widehat{\theta}_t^{\text{PPI}}$. Thus, we propose a similar optimizing strategy as inspired by [1]: at time $t \geqslant 1$, given the obtained $\{\widehat{\lambda}_i\}_{i=1}^{t-1}$ and $\{\widehat{\theta}_i^{\text{PPI}}\}_{i=1}^{t-1}$, we choose an arbitrary $\tilde{\lambda}$, to obtain $\widehat{\theta}_t^{\text{PPI}}(\tilde{\lambda})$ as a temporary surrogate [3]. Then, we further obtain
$$\widehat{\lambda}_t = \arg\min_\lambda F\left(\widehat{V}_t^{\text{PPI}}(\{\widehat{\lambda}_j, \widehat{\theta}_j^{\text{PPI}}(\widehat{\lambda}_j)\}_{j=1}^{t-1}, \lambda, \widehat{\theta}_t^{\text{PPI}}(\tilde{\lambda}); \frac{n}{N})\right),$$

---

[3]Notice that $\widehat{\theta}_t^{\text{PPI}}(\tilde{\lambda})$ is still a consistent estimator of $\theta_t$

where $\widehat{V}_t^{\mathrm{PPI}}$ is obtained via replacing $\nabla_\theta G(\theta_k)$ with their estimation in $V_t^{\mathrm{PPI}}$. In particular, if $F$ satisfies $F(U+V) = F(U) + F(V)$ as our examples of $F(V_t) = \mathrm{Tr}(V_t)$ and $F(V_t) = \mathbf{1}^\top V_t \mathbf{1}$, then we only need to optimize $F\left(\Sigma_{\lambda, \widehat{\theta}_{t-1}^{\mathrm{PPI}}(\widehat{\lambda}_{t-1})}(\widehat{\theta}_t(\tilde{\lambda}); \frac{n}{N})\right)$

To sum up, by the above process, we have the following corollary.

**Corollary 4.2.** *Under Assumption 3.1, A.2, A.3, A.4, and A.5, if $\varepsilon < \frac{\gamma}{\beta}$ and $\frac{n}{N} \to r$ for some $r \geqslant 0$, then for any given $T \geqslant 0$, we have that for all $t \in [T]$,*

$$\left(\widehat{V}_t^{PPI}\left(\{\widehat{\lambda}_j, \widehat{\theta}_j^{PPI}(\widehat{\lambda}_j)\}_{j=1}^t; \frac{n}{N}\right)\right)^{-\frac{1}{2}} \sqrt{n}\left(\widehat{\theta}_t^{PPI}(\widehat{\lambda}_t) - \theta_t\right) \xrightarrow{D} \mathcal{N}(0, I_d).$$

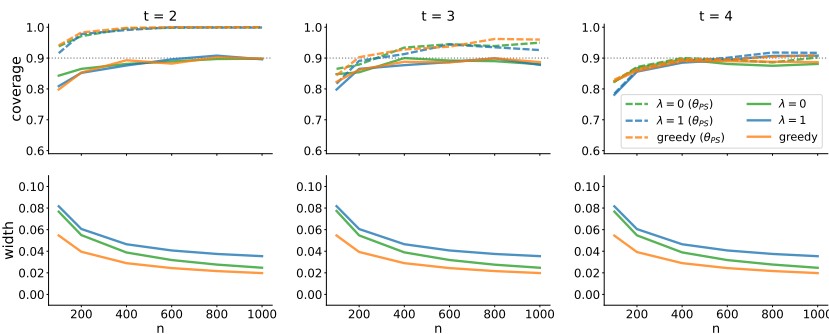

Figure 1: Confidence-region coverage (top row) and width (bottom row) with different choices of $\lambda$. The left, middle, and right columns correspond to inference steps $t = 2$, $t = 3$, and $t = 4$, respectively. The solid and dashed curves correspond to the confidence-region coverage for $\theta_t$ and $\theta_{\mathrm{PS}}$, respectively.

## 5   Experiment

In this section, we further provide numerical experimental results to support our previous theory. A case study on a semi-synthetic dataset is provided in Section B.3.

**Experimental setting.** We follow [20] to construct simulation studies on a performative linear regression problem. Given a parameter $\theta \in \mathbb{R}^d$, data are sampled from $D(\theta)$ as

$$y = \alpha^\top x + \mu^\top \theta + \nu, \ x \sim \mathcal{N}(\mu_x, \Sigma_x), \ \nu \sim \mathcal{N}(0, \sigma_y^2).$$

The distribution map $D(\theta)$ is $\varepsilon$-sensitive with $\varepsilon = \|\mu\|_2$. For unlabeled $x_i^u$, the annotating model is defined as $f(x_i^u) = \alpha^\top x_i^u + \mu^\top \theta + \nu_i, \nu_i \sim \mathcal{N}(-0.2, \sigma_y^2)$. We use the ridge squared loss to measure the performance and update $\theta$: $\ell((x, y); \theta) = \frac{1}{2}(y - \theta^\top x)^2 + \frac{\gamma}{2}\|\theta\|^2$. For easier calculation for smoothness parameter, we truncate the the distribution of $(x, y)$ in our experiment to deal with truncated normal distributions, but this is not necessary in many other choices of updating rules. In the following experiments, we set $d = 2$, $\varepsilon \approx 0.02$, $\gamma = 2$, and $\sigma_y^2 = 0.2$. We set $N = 2000$ and vary the labeled sample size $n$.

**Simulation results for PPI under performativity.** To quantify the results of PPI under performativity, we evaluate the confidence-region coverage and width for three strategies: $\lambda = 0$ (only labeled data), $\lambda = 1$ (full unlabeled data weight), and our optimization method $\lambda = \widehat{\lambda}_t$ as defined in Eq.3. We vary the labeled sample size $n$ and perform $t \in \{2, 3, 4\}$ repeated risk minimization steps, averaging results over 1000 independent trials. In Figure 1, we can find that all three methods approach 0.9 coverage as $n$ grows, while our optimized $\widehat{\lambda}_t$ (orange) achieves the narrowest interval width, supporting its effectiveness to enhance the performative inference. The dashed curves denote the bias-adjusted confidence regions for the performative stable point $\theta_{\mathrm{PS}}$. It can be observed that $\theta_{\mathrm{PS}}$ coverages upper-bound that of $\theta_t$ (solid curves) across steps $t$, and the gap between them vanishes as $t$ grows. This observation verifies the validity of Corollary 3.7.

**Verifying central limit theorem.** To validate the central limit theorem, we sample different $\widehat{\theta}_t$ (here $t = 4$ and $n = 1000$) and visualize the distribution of $\widehat{V}_t^{-1/2}\sqrt{n}(\widehat{\theta}_t - \theta_t)$. We plot the density map

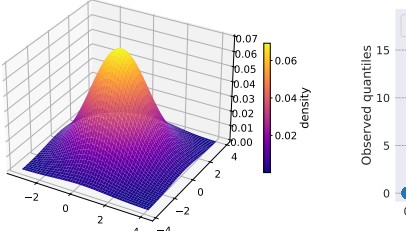
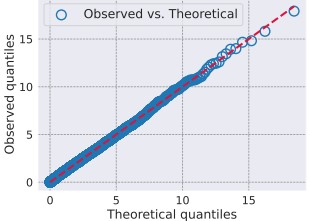

(a) Density map of observed distribution.

(b) Multivariate Q–Q Plot for normality test.

Figure 2: Visualizations to verify the Central Limit Theorem. (a) plots the density map of sampled $\widehat{V}_t^{-1/2}\sqrt{n}(\widehat{\theta}_t - \theta_t)$, while (b) compares the observed distribution with theoretical one $\mathcal{N}(0, I_d)$.

in Figure 2a and find it is close to a normal distribution. In Figure 2b, we further do a normality test with the multivariate Q-Q plot of observed squared Mahalanobis distance over theoretical Chisq quantiles. The tight alignment of points along the identity line (red dashed) verifies that the observed distribution is well-approximated by its asymptotic CLT.

**Estimation results of score matching.** We consider two implementations of the gradient-free score matching estimator $M(z, \theta; \psi)$:

(a). Gaussian parametric: we assume $p(z, \theta) = \mathcal{N}(\mu_p, \Sigma_p)$ and parameterize $\psi = \{\mu_p, \Sigma_p\}$;

(b). DNN-based: a small deep neural network with two hidden layers of width 128.

We collect $\widehat{\theta}_{1:t}$ trajectory and corresponding data $\{z_{1:t,i}\}_{i=1}^{n}$ to train both models via the SGD optimizer with a learning rate of 0.1 to minimize the empirical score-matching objective $J(\psi)$.

In Figure 3, we evaluate the estimation quality of two models by their final training loss $J(\psi)$ and the estimated variance error $\|\widehat{V}_t - V_t\|$ over varying $n$ and $t$. We sample 1000 independent trajectories of $\widehat{\theta}_{1:t}$ and report $J(\psi)$ as averaged $J(\widehat{\theta}; \psi)$ over all $\widehat{\theta}$ in the collection of trajectories. In all settings, both estimators achieve $J(\psi) < 0.05$, indicating the perfect approximation of our learned model $M(z, \theta; \psi)$ to the true $p(z, \theta)$. Correspondingly, the variance-estimation error remains negligible and decreases as $n$ grows, verifying the feasibility of using our score matching models to fit $\nabla_\theta \log p(z, \theta)$ for estimating $\widehat{\nabla G(\theta_k)}$ and $V_t$.

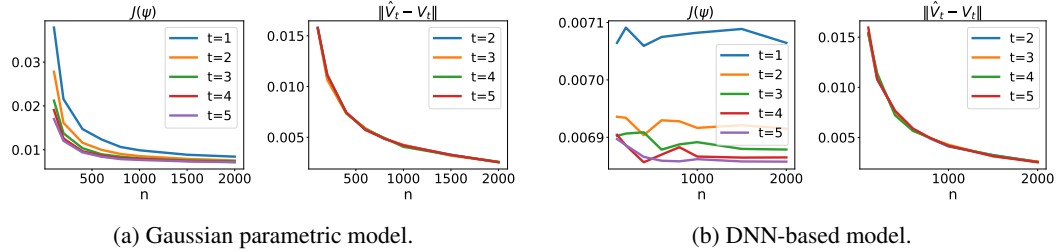

(a) Gaussian parametric model.

(b) DNN-based model.

Figure 3: Evaluating the estimation quality of two designed score matching models.

## 6 Conclusion

In this paper, we introduce an important topic: statistical inference under performativity. We derive results on asymptotic distributions for a widely used iterative process for updating parameters in performative prediction. We further leverage and extend prediction-powered inference to the dynamic setting under performativity. Currently, our framework uses bias-aware inference for $\theta_{\text{PS}}$. Obtaining direct inference methods will be of future interest. Our work can serve as an important tool and guideline for policy-making in a wide range of areas such as social science and economics.

## 7 Acknowledgements

We would like to acknowledge the helpful discussion and suggestions from Tijana Zrinc, Juan Perdomo, Anastasios Angelopoulos, and Steven Wu.

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

# A Theoretical Details

We will include omitted technical details in the main context. We first summarize all the required additional assumptions in Section A.1. Then, we provide omitted proofs for Section 3 in Section A.2 and omitted proofs for Section 4 in Section A.4.

## A.1 Assumptions

In this subsection, we summarize all the additional assumptions we will use to build various theoretical results in this paper. Before we state the assumptions, we need some further notations.

Let us denote $\Psi(\theta)$ as the collection of minimizers of $\int p(z,\theta)\|\nabla_\theta \log p(z,\theta) - s(z,\theta;\psi)\|^2 dz$ for the given $\theta$. We further denote the empirical estimation function for the two terms, i.e.,

$$\int p(z,\theta)\Big(\|s(z,\theta;\psi)\|^2 - 2\nabla_\theta \log p(z,\theta)^\top s(z,\theta;\psi)\Big)dz,$$

as $\widehat{J}_{n,k}(\psi;\theta)$ following the methods in Section 3.2 for any $\theta$ in the trajectory $\{\widehat{\theta}_t\}_{t=1}^T$, where $n$ and $k$ are the number of samples we get at each iteration for $\widehat{\theta}_t$ and perturbed policies.

**Meaning of each assumption.**    Assumption A.1 is used to establish the asymptotic normality of our estimator under performativity. Additionally, we use the fact that the population loss Hessians $H_{\tilde\theta}(\theta) = \nabla_\theta^2 \mathbb{E}_{z\sim\mathcal{D}(\tilde\theta)}\ell(z;\theta)$ are positive definite, which is guaranteed by the strong convexity of the loss function (Assumption 3.1). Assumption A.2 and A.3 are used in the analysis of score matching. Assumption A.2, based on the differentiation lemma [18], ensures the interchangeability of integration and differentiation. Assumption A.3 guarantees the consistency of the score matching estimator. Assumption A.4 and A.5 parallel to Assumption 3.1(a) and A.1, and are used to establish the consistency and asymptotic normality of our PPI estimator under performativity. Conditions such as local Lipschitz continuity and positive definiteness are standard for establishing asymptotic normality. Similar assumptions are also imposed in [1].

**Assumption A.1** (Positive Definiteness & Regularity Conditions for the Estimator). We assume the following.

(a). The loss function satisfies the gradient covariance matrix is uniformly bounded below:

$$V_{\tilde\theta}(\theta) = \mathrm{Cov}_{z\sim\mathcal{D}(\tilde\theta)}\big(\nabla_\theta \ell(z;\theta)\big) \succeq cI,$$

for any $\tilde\theta$, where $c > 0$ is a constant.

(b). For any sample size $n$, assume the M-estimator $\widehat{\theta}_t$ has a density function with respect to the Lebesgue measure, and its characteristic function is absolutely integrable.

**Assumption A.2** (Regularity Condition for $M$). Assume that for $\forall i$:

(a). The function $z \mapsto p(z,\theta)\dfrac{\partial \log M(z,\theta;\psi)}{\partial \theta^{(i)}}$ is Lebesgue integrable.

(b). For almost every $z \in \mathcal{Z}$ (with respect to Lebesgue measure), the partial derivative

$$\frac{\partial}{\partial\theta^{(i)}}\left[p(z,\theta)\frac{\partial \log M(z,\theta;\psi)}{\partial\theta^{(i)}}\right]$$

exists.

(c). There exists a Lebesgue-integrable function $H(z)$ such that for almost every $z \in \mathcal{Z}$,

$$\left|\frac{\partial}{\partial\theta^{(i)}}\left[p(z,\theta)\frac{\partial \log M(z,\theta;\psi)}{\partial\theta^{(i)}}\right]\right| \leqslant H(z).$$

**Assumption A.3** (Consistency of Optimizer). We let $k$ grows along with $n$ such that $n \to \infty$ leads to $k \to \infty$. We assume that the class $M(z,\theta;\psi)$ is rich enough that for all $\theta \in \Theta$, there exists $\psi^*(\theta)$ such that $M(z,\theta;\psi^*(\theta)) = p(z,\theta)$. Moreover, for the underlying trajectory $\{\theta_t\}_{t=1}^T$,

$$\lim_{n\to\infty} \arg\min_\psi \widehat{J}_{n,k}(\psi;\widehat{\theta}_t) \subseteq \Psi(\theta_t).$$

**Assumption A.4** (Local Lipschitzness with $f$). Loss function $\ell(x, f(x); \theta)$ is locally Lipschitz: for each $\theta \in \Theta$, there exist a neighborhood $\Upsilon(\theta)$ of $\theta$ such that $\ell(x, f(x); \tilde{\theta})$ is $L^f(x)$ Lipschitz w.r.t $\tilde{\theta}$ for all $\tilde{\theta} \in \Upsilon(\theta)$ and $\mathbb{E}_{x \sim \mathcal{D}_{\mathcal{X}}(\theta)} L^f(x) < \infty$.

**Assumption A.5** (Positive Definiteness with $f$ & Regularity Conditions for the PPI Estimator). We assume the following.

(a). Assume the loss function satisfies the the gradient covariance matrices are uniformly bounded below:

$$V_{\tilde{\theta}}(\theta) = \mathrm{Cov}_{z \sim \mathcal{D}(\tilde{\theta})}\big(\nabla_\theta \ell(z; \theta)\big) \succeq cI, \quad V_{\tilde{\theta}}^f(\theta) = \mathrm{Cov}_{x \sim \mathcal{D}_{\mathcal{X}}(\tilde{\theta})}(\nabla_\theta \ell(x, f(x); \theta)) \succeq cI,$$

for any $\tilde{\theta}, \theta$, where $c > 0$ is a constant.

(b). For any sample size $n$, assume $\widehat{\theta}_t^{\mathrm{PPI}}$ has a density function with respect to the Lebesgue measure, and its characteristic function is absolutely integrable.

## A.2 Details of Section 3: Theory of Inference under Performativity

We provide the omitted details in Section 3.

### A.2.1 Consistency and Central Limit Theorem of $\widehat{\theta}_t$

Let us denote:

$$\mathcal{L}_{\tilde{\theta}}(\theta) := \mathbb{E}_{z \sim \mathcal{D}(\tilde{\theta})} \ell(z; \theta), \quad \mathcal{L}_{\tilde{\theta}, n}(\theta) := \frac{1}{n} \sum_{i=1}^n \ell(z_i; \theta),$$

where the samples $z_i = (x_i, y_i) \sim \mathcal{D}(\tilde{\theta})$ are drawn from the distribution under $\tilde{\theta}$.

**Proposition A.6** (Consistency of $\widehat{\theta}_t$, Restatement of Proposition 3.3). *Under Assumption 3.1, if $\varepsilon < \frac{\gamma}{\beta}$, then for any given $T \geqslant 0$, we have that for all $t \in [T]$,*

$$\widehat{\theta}_t \xrightarrow{P} \theta_t.$$

*Proof.* Let us denote $\widehat{G}(\theta) := \mathrm{argmin}_{\theta' \in \Theta} \frac{1}{n} \sum_{i=1}^n \ell(z_i; \theta')$ where the samples $z_i \sim \mathcal{D}(\theta)$ are drawn for some parameter $\theta$ along the dynamic trajectory $\theta_0 \to \widehat{\theta}_1 \to \cdots \widehat{\theta}_t \to \cdots$.

$$
\begin{aligned}
\|\theta_t - \widehat{\theta}_t\| &= \|G(\theta_{t-1}) - \widehat{G}(\widehat{\theta}_{t-1})\| \\
&\leqslant \|G(\widehat{\theta}_{t-1}) - \widehat{G}(\widehat{\theta}_{t-1})\| + \|G(\theta_{t-1}) - G(\widehat{\theta}_{t-1})\| \\
&\leqslant \|G(\widehat{\theta}_{t-1}) - \widehat{G}(\widehat{\theta}_{t-1})\| + \varepsilon \frac{\beta}{\gamma} \|\theta_{t-1} - \widehat{\theta}_{t-1}\|,
\end{aligned}
$$

where the last inequality follows from the results derived by [25], under Assumption 3.1, we have $\|G(\theta) - G(\theta')\| \leqslant \frac{\varepsilon \beta}{\gamma} \|\theta - \theta'\|$.

Notice that $\mathbb{E}(\mathcal{L}_{\widehat{\theta}_{t-1}, n}(\theta)) = \mathcal{L}_{\widehat{\theta}_{t-1}}(\theta)$. By local Lipschitz condition, there exists $\varepsilon_0 > 0$ such that

$$\sup_{\theta: \|\theta - G(\widehat{\theta}_{t-1})\| \leqslant \varepsilon_0} |\mathcal{L}_{\widehat{\theta}_{t-1}, n}(\theta) - \mathcal{L}_{\widehat{\theta}_{t-1}}(\theta)| \xrightarrow{P} 0.$$

Since $\ell$ is strongly convex for any $\theta$, $G(\widehat{\theta}_{t-1})$ is unique. Then we know that there exists $\delta$ such that $\mathcal{L}_{\widehat{\theta}_{t-1}, n}(\theta) - \mathcal{L}_{\widehat{\theta}_{t-1}}(G(\widehat{\theta}_{t-1})) > \delta$ for all $\theta$ in $\{\theta \mid \|\theta - G(\widehat{\theta}_{t-1})\| = \varepsilon_0\}$. Then it follows that:

$$
\begin{aligned}
&\inf_{\|\theta - G(\widehat{\theta}_{t-1})\| = \varepsilon_0} \mathcal{L}_{\widehat{\theta}_{t-1}, n}(\theta) - \mathcal{L}_{\widehat{\theta}_{t-1}, n}(G(\widehat{\theta}_{t-1})) \\
&= \inf_{\|\theta - G(\widehat{\theta}_{t-1})\| = \varepsilon_0} \Big( (\mathcal{L}_{\widehat{\theta}_{t-1}, n}(\theta) - \mathcal{L}_{\widehat{\theta}_{t-1}}(\theta)) + (\mathcal{L}_{\widehat{\theta}_{t-1}}(\theta) - \mathcal{L}_{\widehat{\theta}_{t-1}}(G(\widehat{\theta}_{t-1}))) \\
&\quad + (\mathcal{L}_{\widehat{\theta}_{t-1}}(G(\widehat{\theta}_{t-1})) - \mathcal{L}_{\widehat{\theta}_{t-1}, n}(G(\widehat{\theta}_{t-1}))) \Big)
\end{aligned}
$$

$$\geqslant \delta - o_P(1).$$

Then we consider any fixed $\theta$ such that $\|\theta - G(\widehat{\theta}_{t-1})\| \geqslant \varepsilon_0$ it follows that

$$\mathcal{L}_{\widehat{\theta}_{t-1},n}(\theta) - \mathcal{L}_{\widehat{\theta}_{t-1},n}(G(\widehat{\theta}_{t-1})) \geqslant \frac{\theta - G(\widehat{\theta}_{t-1})}{\omega - G(\widehat{\theta}_{t-1})} \left( \mathcal{L}_{\widehat{\theta}_{t-1},n}(\omega) - \mathcal{L}_{\widehat{\theta}_{t-1},n}(G(\widehat{\theta}_{t-1})) \right)$$

$$\geqslant \frac{\|\theta - G(\widehat{\theta}_{t-1})\|}{\varepsilon_0}(\delta - o_P(1)) \geqslant \delta - o_P(1),$$

where the first inequality holds for any $\omega$ by the convexity condition of $\mathcal{L}_{\widehat{\theta}_{t-1},n}(\theta)$, and the second inequality holds as we take $\omega = \frac{\theta - G(\widehat{\theta}_{t-1})}{\|\theta - G(\widehat{\theta}_{t-1})\|}\varepsilon_0 + G(\widehat{\theta}_{t-1})$ and using the above result. Thus no $\theta$ such that $\|\theta - G(\widehat{\theta}_{t-1})\| = \varepsilon_0$ can be the minimizer of $\mathcal{L}_{\widehat{\theta}_{t-1},n}(\theta)$. Then $\|G(\widehat{\theta}_{t-1}) - \widehat{G}(\widehat{\theta}_{t-1})\| \xrightarrow{P} 0$.

We then have, for a given $T \geqslant 0$, we have that for all $t \in [T]$,

$$\|\widehat{\theta}_t - \theta_t\| \leqslant \sum_{i=0}^{t} (\varepsilon \frac{\beta}{\gamma})^{t-i} \|G(\widehat{\theta}_i) - \widehat{G}(\widehat{\theta}_i)\| \xrightarrow{P} 0.$$

Thus, we conclude that $\widehat{\theta}_t \xrightarrow{P} \theta_t$. ∎

**Theorem A.7** (Central Limit Theorem of $\widehat{\theta}_t$, Restatement of Theorem 3.4). *Under Assumption 3.1 and A.1, if $\varepsilon < \frac{\gamma}{\beta}$, then for any given $T \geqslant 0$, we have that for all $t \in [T]$,*

$$\sqrt{n}(\widehat{\theta}_t - \theta_t) \xrightarrow{D} \mathcal{N}(0, V_t)$$

*with*

$$V_t = \sum_{i=1}^{t} \left[ \prod_{k=i}^{t-1} \nabla G(\theta_k) \right] \Sigma_{\theta_{i-1}}(\theta_i) \left[ \prod_{k=i}^{t-1} \nabla G(\theta_k) \right]^{\top}.$$

*In particular, $\nabla G(\theta_k) = -H_{\theta_k}(\theta_{k+1})^{-1} \left( \nabla_{\tilde{\theta}} \mathbb{E}_{z \sim \mathcal{D}(\theta_k)} \nabla_\theta \ell(z; \theta_{k+1}) \right)$, where $\nabla_{\tilde{\theta}}$ is taking gradient for the parameter in $\mathcal{D}(\tilde{\theta})$, $\nabla_\theta$ is taking gradient for the parameter in $\ell(z; \theta)$ and $\prod_{k=t}^{t-1} \nabla G(\theta_k) = I_d$.*

*Proof.* Let $U_t := \sqrt{n}(\widehat{\theta}_t - \theta_t)$ and denote $\tilde{\theta}_t = G(\widehat{\theta}_{t-1})$. We make the following decomposition:

$$\widehat{\theta}_t - \theta_t = \underbrace{(\tilde{\theta}_t - \theta_t)}_{(1)} + \underbrace{(\widehat{\theta}_t - \tilde{\theta}_t)}_{(2)}.$$

**Step 1: Conditional distribution of $U_t | U_{t-1}$.**

For term (1), we have
$$\sqrt{n}(\tilde{\theta}_t - \theta_t) = \sqrt{n}(G(\widehat{\theta}_{t-1}) - G(\theta_{t-1})).$$

For term (2), the empirical process analysis in [1] establishes that

$$\sqrt{n}(\widehat{\theta}_t - \tilde{\theta}_t) \mid \widehat{\theta}_{t-1} \xrightarrow{D} \mathcal{N}(0, \Sigma_{\widehat{\theta}_{t-1}}(\tilde{\theta}_t)),$$

where the variance is given by

$$\Sigma_{\widehat{\theta}_{t-1}}(\tilde{\theta}_t) = H_{\widehat{\theta}_{t-1}}(\tilde{\theta}_t)^{-1} V_{\widehat{\theta}_{t-1}}(\tilde{\theta}_t) H_{\widehat{\theta}_{t-1}}(\tilde{\theta}_t)^{-1}.$$

Conditioning on $\widehat{\theta}_{t-1}$ and considering the distribution $D(\widehat{\theta}_{t-1})$, for any function $h$, we use the following shorthand notations:

$$\mathbb{E}_n h := \frac{1}{n} \sum_{i=1}^{n} h(x_i, y_i), \quad \mathbb{G}_n h := \sqrt{n}(\mathbb{E}_n h - \mathbb{E}_{(x,y) \sim \mathcal{D}(\widehat{\theta}_{t-1})}[h(x, y)]).$$

Note that $\tilde{\theta}_t = G(\widehat{\theta}_{t-1})$. Recall that

$$\mathcal{L}_{\tilde{\theta}}(\theta) := \mathbb{E}_{(x,y)\sim\mathcal{D}(\tilde{\theta})}\ell(x,y;\theta), \quad \mathcal{L}_{\tilde{\theta},n} := \frac{1}{n}\sum_{i=1}^n \ell(x_i,y_i;\theta), \text{ where } (x_i,y_i) \sim \mathcal{D}(\tilde{\theta}).$$

Under the assumptions, Lemma 19.31 in [31] implies that for every sequence $h_n = O_P(1)$, we have

$$\mathbb{G}_n\left[\sqrt{n}\left(\ell(x,y;\tilde{\theta}_t + \frac{h_n}{\sqrt{n}}) - \ell(x,y;\tilde{\theta}_t)\right) - h_n^\top \nabla_\theta \ell(x,y;\tilde{\theta}_t)\right] \xrightarrow{P} 0.$$

Applying second-order Taylor expansion, we obtain that

$$n\mathbb{E}_n\left(\ell(x,y;\tilde{\theta}_t + \frac{h_n}{\sqrt{n}}) - \ell(x,y;\tilde{\theta}_t)\right) = n\left(\mathcal{L}_{\widehat{\theta}_{t-1}}(\tilde{\theta}_t + \frac{h_n}{\sqrt{n}}) - \mathcal{L}_{\widehat{\theta}_{t-1}}(\tilde{\theta}_t)\right)$$
$$+ h_n^\top \mathbb{G}_n \nabla_\theta \ell(x,y;\tilde{\theta}_t) + o_p(1)$$
$$= \frac{1}{2}h_n^\top H_{\widehat{\theta}_{t-1}}(\tilde{\theta}_t)h_n + h_n^\top \mathbb{G}_n \nabla_\theta \ell(x,y;\tilde{\theta}_t) + o_p(1).$$

Set $h_n^* = \sqrt{n}(\widehat{\theta}_t - \tilde{\theta}_t)$ and $h_n = -H_{\widehat{\theta}_{t-1}}(\tilde{\theta}_t)^{-1}\mathbb{G}_n\nabla_\theta\ell(x,y;\tilde{\theta}_t)$, Corollary 5.53 in [31] implies they are $O_P(1)$.

Since $\widehat{\theta}_t$ is the minimizer of $\mathcal{L}_{n,\widehat{\theta}_{t-1}}$, the first term is smaller than the second term. We can rearrange the terms and obtain:
$$\frac{1}{2}(h_n^* - h_n)^T H_{\widehat{\theta}_{t-1}}(\tilde{\theta}_t)(h_n^* - h_n) = o_P(1),$$

which leads to $h_n^* - h_n = O_P(1)$. Then the above asymptotic normality result follows directly by applying the central limit theorem (CLT) to the following terms, conditioning on $\widehat{\theta}_{t-1}$:

$$\sqrt{n}(\widehat{\theta}_t - \tilde{\theta}_t) \mid \widehat{\theta}_{t-1} = -H_{\widehat{\theta}_{t-1}}(\tilde{\theta}_t)^{-1}S + o_P(1),$$
$$S = \sqrt{\frac{1}{n}\sum_{i=1}^n\left(\nabla_\theta\ell(x_{t,i},y_{t,i};\tilde{\theta}_t) - \mathbb{E}_{(x,y)\sim\mathcal{D}(\widehat{\theta}_{t-1})}[\nabla_\theta\ell(x,y;\tilde{\theta}_t)]\right)}.$$

Note that, conditioning on $\widehat{\theta}_{t-1}$, (1) is a constant. Therefore, (1) and (2) follow a joint Gaussian distribution. Consequently, given $U_{t-1}$, the conditional distribution of $U_t$ is given by:

$$U_t \mid U_{t-1} = \sqrt{n}(\widehat{\theta}_t - \theta_t) \mid \widehat{\theta}_{t-1}$$
$$= \sqrt{n}(\tilde{\theta}_t - \theta_t) + \sqrt{n}(\widehat{\theta}_t - \tilde{\theta}_t) \mid \widehat{\theta}_{t-1}$$
$$= \sqrt{n}(G(\widehat{\theta}_{t-1}) - G(\theta_{t-1})) + \sqrt{n}(\widehat{\theta}_t - \tilde{\theta}_t) \mid \widehat{\theta}_{t-1}$$
$$\xrightarrow{D} \mathcal{N}\left(\sqrt{n}(G(\widehat{\theta}_{t-1}) - G(\theta_{t-1})), \Sigma_{\widehat{\theta}_{t-1}}(\tilde{\theta}_t)\right).$$
$$= \mathcal{N}\left(\sqrt{n}(G(\frac{U_{t-1}}{\sqrt{n}} + \theta_{t-1}) - G(\theta_{t-1})), \Sigma_{\frac{U_{t-1}}{\sqrt{n}}+\theta_{t-1}}(G(\frac{U_{t-1}}{\sqrt{n}} + \theta_{t-1}))\right).$$

For later references, we denote $U_t \mid U_{t-1} \xrightarrow{D} \mathcal{N}(\mu(U_{t-1}), \Sigma(U_{t-1}))$.

**Step 2: Marginal distribution of $U_t$.** We calculate the characteristic function of $U_t$ by induction. To begin with, we directly have

$$X_1 \xrightarrow{D} \mathcal{N}(0, V_1), \quad V_1 = \Sigma_{\theta_0}(\theta_1).$$

Now, assume that $U_{t-1} \xrightarrow{D} \mathcal{N}(0, V_{t-1})$, we derive the joint distribution of $(U_t, U_{t-1})$ and marginal distribution of $U_t$. Then we have, the characteristics functions $\phi$ and the probability density function $p$ of distributions $U_{t-1}$ and $U_t \mid U_{t-1}$ follow:

$$\phi_{U_{t-1}}(s) \to \phi_{\mathcal{N}(0,V_{t-1})}(s) = \exp(-\frac{1}{2}s^T V_{t-1}s), \quad p_{U_{t-1}}(u) = \frac{1}{(2\pi)^d}\int e^{-iz^T u}\phi_{U_{t-1}}(z)dz,$$

$$\phi_{U_t|U_{t-1}}(s) \to \phi_{\mathcal{N}(\mu(U_{t-1}),\Sigma(U_{t-1}))}(s) = \exp(is^T\mu(U_{t-1}) - \frac{1}{2}s^T\Sigma(U_{t-1})s).$$

Then we have

$$\phi_{U_t}(s) = \mathbb{E}e^{is^TU_t} = \mathbb{E}\big(\mathbb{E}(e^{is^\top U_t} \mid U_{t-1})\big) = E_{U_{t-1}}\phi_{U_t|U_{t-1}}(s \mid U_{t-1})$$

$$= \int \phi_{U_t|U_{t-1}}(s \mid u)\, p_{U_{t-1}}(u)\, du$$

$$= \int \phi_{U_t|U_{t-1}}(s \mid u)\, \frac{1}{(2\pi)^d} \int e^{-iz^\top u}\, \phi_{U_{t-1}}(z)\, dz\, du$$

$$= \frac{1}{(2\pi)^d} \iint \phi_{U_t|U_{t-1}}(s \mid u)\, \phi_{U_{t-1}}(z)\, e^{-iz^\top u}\, dz\, du$$

$$= \frac{1}{(2\pi)^d} \iint \exp\big(is^\top\mu(U_{t-1}) - \tfrac{1}{2}s^\top\Sigma(U_{t-1})s\big)\exp\big(-\tfrac{1}{2}z^\top V_{t-1}z\big) e^{-iz^\top u}\, dz\, du$$

$$= \frac{1}{(2\pi)^d} \int \exp\big(is^\top\mu(U_{t-1}) - \tfrac{1}{2}s^\top\Sigma(U_{t-1})s\big)\Big(\int \exp\big(-\tfrac{1}{2}z^\top V_{t-1}z - iz^\top u\big)\, dz\Big)\, du$$

$$= \frac{1}{(2\pi)^d} \int \exp\big(is^\top\mu(U_{t-1}) - \tfrac{1}{2}s^\top\Sigma(U_{t-1})s\big)$$

$$\times \Big(\int \exp\big(-\tfrac{1}{2}u^\top V_{t-1}^{-1}u\big)\exp\big(-\tfrac{1}{2}(z - V_{t-1}^{-1}iu)^\top V_{t-1}(z - V_{t-1}^{-1}iu)\big)\, dz\Big)du$$

$$= \frac{1}{(2\pi)^d} \int \exp\big(is^\top\mu(U_{t-1}) - \tfrac{1}{2}s^\top\Sigma(U_{t-1})s\big)\big((2\pi)^{\frac{d}{2}}\frac{1}{\det|V_{t-1}|}\cdot\exp(-\frac{1}{2}u^\top V_{t-1}^{-1}u)\big)\, du$$

$$= \frac{1}{(2\pi)^{\frac{d}{2}}\det|V_{t-1}|} \int \exp(is^T\mu(U_{t-1}) - \frac{1}{2}s^T\Sigma(U_{t-1})s - \frac{1}{2}u^T V_{t-1}u)du.$$

Apply dominant convergence theorem to $\lim_{n\to\infty}\phi_{U_t}(s)$, we have:

$$\lim_{n\to\infty}\phi_{U_t}(s) = \lim_{n\to\infty}\frac{1}{(2\pi)^{\frac{d}{2}}\det|V_{t-1}|}\int \exp(is^T\mu(U_{t-1}) - \frac{1}{2}s^T\Sigma(U_{t-1})s - \frac{1}{2}u^T V_{t-1}u)du$$

$$= \lim_{n\to\infty}\frac{1}{(2\pi)^{\frac{d}{2}}\det|V_{t-1}|}\int \exp(is^T\sqrt{n}(G(\frac{U_{t-1}}{\sqrt{n}} + \theta_{t-1}) - G(\theta_{t-1}))$$

$$- \frac{1}{2}s^T\Sigma_{\frac{U_{t-1}}{\sqrt{n}}+\theta_{t-1}}(G(\frac{U_{t-1}}{\sqrt{n}} + \theta_{t-1}))s - \frac{1}{2}u^T V_{t-1}u)du$$

$$= \frac{1}{(2\pi)^{\frac{d}{2}}\det|V_{t-1}|}\int \lim_{n\to\infty}\exp(is^T\sqrt{n}(G(\frac{U_{t-1}}{\sqrt{n}} + \theta_{t-1}) - G(\theta_{t-1}))$$

$$- \frac{1}{2}s^T\Sigma_{\frac{U_{t-1}}{\sqrt{n}}+\theta_{t-1}}(G(\frac{U_{t-1}}{\sqrt{n}} + \theta_{t-1}))s - \frac{1}{2}u^T V_{t-1}u)du$$

$$= \frac{1}{(2\pi)^{\frac{d}{2}}\det|V_{t-1}|}\int \exp(is^T\nabla G(\theta_{t-1})u - \frac{1}{2}s^T\Sigma_{\theta_{t-1}}(G(\theta_{t-1}))s - \frac{1}{2}u^T V_{t-1}u)du$$

$$= \exp(-\frac{1}{2}s^T\nabla G(\theta_{t-1})V_{t+1}\nabla G(\theta_{t-1})^T s - \frac{1}{2}s^T\Sigma_{U_{t-1}}(\theta_t)s),$$

which is the characteristic function of $\mathcal{N}(0, V_t)$, where $V_t = \nabla G(\theta_{t-1})V_{t-1}\nabla G(\theta_{t-1})^\top + \Sigma_{\theta_{t-1}}(\theta_t)$. Here we use the fact that $\lim_{n\to\infty}\sqrt{n}\big(G(\frac{y}{\sqrt{n}} + \theta_{t-1}) - G(\theta_{t-1})\big) = \nabla G(\theta_{t-1})y$, and the dominant convergence theorem holds as we have

$$|\exp(is^T\sqrt{n}(G(\frac{U_{t-1}}{\sqrt{n}} + \theta_{t-1}) - G(\theta_{t-1}))$$

$$- \frac{1}{2}s^T\Sigma_{\frac{U_{t-1}}{\sqrt{n}}+\theta_{t-1}}(G(\frac{U_{t-1}}{\sqrt{n}} + \theta_{t-1}))s - \frac{1}{2}u^T V_{t-1}u| \leqslant |\exp(-\frac{1}{2}u^T V_{t-1}u)|.$$

Thus we conclude by induction that

$$U_t \xrightarrow{D} \mathcal{N}(0, V_t),$$

$$V_t = \sum_{i=1}^{t} \left[ \prod_{k=i}^{t-1} \nabla G(\theta_k) \right] \Sigma_{\theta_{i-1}}(\theta_i) \left[ \prod_{k=i}^{t-1} \nabla G(\theta_k) \right]^\top.$$

And by the theorem of implicit function, we can calculate the gradient of $G$ as follows

$$\nabla G(\theta) = - \left[ (\nabla^2_\psi \mathop{\mathbb{E}}_{(x,y)\sim\mathcal{D}(\theta)} \ell(x,y;\psi))|_{\psi=G(\theta)} \right]^{-1} \left( \nabla_\psi \nabla_{\tilde{\theta}} \mathop{\mathbb{E}}_{(x,y)\sim\mathcal{D}(\theta)} \ell(x,y;\psi) \right)|_{\psi=G(\theta)}.$$

$$\begin{aligned}
\nabla G(\theta_k) &= - \left[ \mathop{\mathbb{E}}_{(x,y)\sim\mathcal{D}(\theta_k)} \nabla^2_\theta \ell(x,y;\theta_{k+1}) \right]^{-1} \left( \nabla_{\tilde{\theta}} \mathop{\mathbb{E}}_{(x,y)\sim\mathcal{D}(\theta_k)} \nabla_\theta \ell(x,y;\theta_{k+1}) \right) \\
&= - H_{\theta_k}(\theta_{k+1})^{-1} \left( \nabla_{\tilde{\theta}} \mathop{\mathbb{E}}_{(x,y)\sim\mathcal{D}(\theta_k)} \nabla_\theta \ell(x,y;\theta_{k+1}) \right) \\
&= - H_{\theta_k}(\theta_{k+1})^{-1} \mathbb{E}_{z\sim\mathcal{D}(\theta_k)} [\nabla_\theta \ell(z,\theta_{k+1}) \nabla_\theta \log p(z,\theta_k)^\top].
\end{aligned}$$

∎

### A.2.2 Score Matching

In this part, we provide details about our score matching mechanism.

Given that

$$\nabla G(\theta_k) = - H_{\theta_k}(\theta_{k+1})^{-1} \mathbb{E}_{z\sim\mathcal{D}(\theta_k)} [\nabla_\theta \ell(z,\theta_{k+1}) \nabla_\theta \log p(z,\theta_k)^\top],$$

once we have a good estimation of $\nabla_\theta \log p(z, \theta_k)$ for all $z \in \mathcal{Z}$, $\nabla G(\theta_k)$ could be easily estimated by samples.

Recall that we use a model $M(z, \theta; \psi)$ parameterized by $\psi$ to approximate $p(z, \theta)$. Inspired by the objective in [14], for any given $\theta$ (e.g., $\widehat{\theta}_t$), we aim to optimize the following objective parameterized by $\psi$:

$$\begin{aligned}
J(\theta; \psi) &= \int p(z, \theta) \| \nabla_\theta \log p(z, \theta) - s(z, \theta; \psi) \|^2 dz \\
&= \int p(z, \theta) \Big( \| \nabla_\theta \log p(z, \theta) \|^2 + \| s(z, \theta; \psi) \|^2 - 2 \nabla_\theta \log p(z, \theta)^\top s(z, \theta; \psi) \Big) dz
\end{aligned}$$

where $s(z, \theta; \psi) = \nabla_\theta \log M(z, \theta; \psi)$.

As mentioned in the main context, the first term is unrelated to $\psi$; the second term involves model $M$ that is chosen by us, so we have the analytical expression of $s(z, \theta; \psi)$. Thus, our key task will be estimating the third term, which involves $\mathcal{K}(\theta; \psi) := \int p(z, \theta) \nabla_\theta \log p(z, \theta)^\top s(z, \theta; \psi) dz$.

**Lemma A.8** (Restatement of Lemma 3.5). *Under Assumption A.2, we have*

$$\mathcal{K}(\theta; \psi) = \sum_{i=1}^{d} \left[ \frac{\partial}{\partial \theta^{(i)}} \int p(z, \theta) \frac{\partial \log M(z, \theta; \psi)}{\partial \theta^{(i)}} dz - \int p(z, \theta) \frac{\partial^2 \log M(z, \theta; \psi)}{\partial \theta^{(i)2}} dz \right]$$

*where $\theta^{(i)}$ is the $i$-th coordinate of $\theta$.*

*Proof.* Recall that $\theta$ is of $d$-dimension.

$$\begin{aligned}
\int p(z, \theta) \nabla_\theta \log p(z, \theta)^\top s(z, \theta; \psi) dz &= \sum_{i=1}^{d} \int p(z, \theta) \frac{\partial \log p(z, \theta)}{\partial \theta^{(i)}} \cdot \frac{\partial \log M(z, \theta; \psi)}{\partial \theta^{(i)}} dz \\
&= \sum_{i=1}^{d} \int p(z, \theta) \frac{\partial \log p(z, \theta)}{\partial \theta^{(i)}} \cdot \frac{\partial \log M(z, \theta; \psi)}{\partial \theta^{(i)}} dz \\
&= \sum_{i=1}^{d} \int \frac{\partial p(z, \theta)}{\partial \theta^{(i)}} \cdot \frac{\partial \log M(z, \theta; \psi)}{\partial \theta^{(i)}} dz.
\end{aligned}$$

Then, we study $\int \frac{\partial p(z,\theta)}{\partial\theta^{(i)}} \cdot \frac{\partial \log M(z,\theta;\psi)}{\partial\theta^{(i)}} dz$. Under Assumption A.2, the integral and differentiation of the following equation is exchangeable, i.e.,

$$\frac{\partial}{\partial\theta^{(i)}} \int p(z,\theta) \frac{\partial M(z,\theta;\psi)}{\partial\theta^{(i)}} dz = \int \frac{p(z,\theta)}{\partial\theta^{(i)}} \frac{\partial M(z,\theta;\psi)}{\partial\theta^{(i)}} dz.$$

According to integral by parts, we have

$$\int \frac{\partial p(z,\theta)}{\partial\theta^{(i)}} \cdot \frac{\partial \log M(z,\theta;\psi)}{\partial\theta^{(i)}} dz = \frac{\partial}{\partial\theta^{(i)}} \int p(z,\theta) \frac{\partial M(z,\theta;\psi)}{\partial\theta^{(i)}} dz - \int p(z,\theta) \frac{\partial^2 \log M(z,\theta;\psi)}{\partial\theta^{(i)2}} dz.$$

Thus, our proof is completed. ∎

The rest of the estimation process via policy perturbation is provided in the main context in Section 3.

The other part omitted in Section 3 is the details about Eq. 2 that

$$\widehat{V}_t^{-1/2} \sqrt{n}(\widehat{\theta}_t - \theta_t) \xrightarrow{D} \mathcal{N}(0, I_d).$$

Here $\widehat{V}_t$ denotes the sample-based estimator of the variance, obtained by plugging in the empirical Hessian and empirical covariance matrices:

$$\widehat{H}_{\widehat{\theta}_{t-1}}(\widehat{\theta}_t) = \widehat{\mathbb{E}}_{z\sim\mathcal{D}(\widehat{\theta}_{t-1})} \nabla_\theta^2 \ell(z;\widehat{\theta}_t), \quad \widehat{\mathrm{Cov}}_{z\sim\mathcal{D}(\widehat{\theta}_{t-1})}\big(\nabla_\theta\ell(z;\widehat{\theta}_t)\big),$$

as well as the estimator for $\nabla G(\widehat{\theta}_{t-1})$:

$$-\widehat{H}_{\widehat{\theta}_{t-1}}(\widehat{\theta}_t)^{-1} \widehat{\mathbb{E}}_{z\sim\mathcal{D}(\widehat{\theta}_{t-1})}[\nabla_\theta\ell(z,\widehat{\theta}_t)\nabla_\theta \log M(z,\widehat{\theta}_{t-1},\widehat{\psi})^\top],$$

where $\widehat{\psi}$ is obtained by minimizing $\widehat{J}_{n,k}$.

Eq. 2 is a direct result following Slutsky's theorem. Assumption A.3 makes sure the empirical optimizer set can converge to the population optimizer set. Then other parts such as estimation of the Hessian matrix etc. could all be directly obtained by standard law of large numbers. Thus, we can directly use Slutsky's theorem to obtain Eq. 2.

### A.3 Policy Perturbation

In this part, we prove the validity of $\widehat{g}_k$ as an estimator of $\nabla G(\theta_k)$. Recall that we have $s(z,\theta;\psi) = \nabla_\theta \log M(z,\theta;\psi)$ and we further denote $s_i = \frac{\partial M(z,\theta;\psi)}{\partial\theta^{(i)}}$, and use $\mathbb{E}_\theta, \widehat{\mathbb{E}}_{\theta,n}$ for the expectation $\mathbb{E}_{z\sim D(\theta)}$ and empirical expectation $\mathbb{E}_{z\sim\widehat{D}_n(\theta)}$ respectively.

Firstly, we define the following function families:

$$\mathcal{F}_{1,\theta} := \{s(\cdot,\theta;\psi) : \psi \in \Psi\},$$
$$\mathcal{F}_{2,\theta}^{(i)} := \{\frac{\partial}{\partial\theta^{(i)}} s_i(\cdot,\theta;\psi) : \psi \in \Psi\},$$
$$\mathcal{F}_{3,\theta}^{(i)} := \{s_i(\cdot,\theta;\psi) : \psi \in \Psi\}.$$

Further, we set

$$\widehat{J}_{n,k}(\theta;\psi) := \widehat{\mathbb{E}}_{\theta,n}[\|s(z,\theta;\psi)\|^2] + \mathbb{E}_\theta\big[\|\nabla_\theta \log p(z,\theta)\|^2\big] + 2\sum_{i=1}^d \big[\widehat{\mathbb{E}}_{\theta,n}[\frac{\partial}{\partial\theta^{(i)}} s_i(z,\theta;\psi)]\big]$$

$$- 2\sum_{i=1}^d \frac{1}{\eta}\bigg(\widehat{\mathbb{E}}_{\theta+\eta e^{(i)},k}\big[s_i(z,\theta+\eta e^{(i)};\psi)\big] - \widehat{\mathbb{E}}_{\theta,n}\big[s_i(z,\theta;\psi)\big]\bigg),$$

$$J_n(\theta;\psi) := \mathbb{E}_\theta[\|s(z,\theta;\psi)\|^2] + \mathbb{E}_\theta\big[\|\nabla_\theta \log p(z,\theta)\|^2\big] + 2\sum_{i=1}^d \big[\mathbb{E}_\theta[\frac{\partial}{\partial\theta^{(i)}} s_i(z,\theta;\psi)]\big]$$

$$- 2\sum_{i=1}^d \frac{1}{\eta}\bigg(\mathbb{E}_{\theta+\eta e^{(i)}}\big[s_i(z,\theta+\eta e^{(i)};\psi)\big] - \mathbb{E}_\theta\big[s_i(z,\theta;\psi)\big]\bigg).$$

**Assumption A.9.** We assume that the score function $s$, distribution map $D$ and the corresponding function families $\{\mathcal{F}_{1,\theta}, \mathcal{F}_{2,\theta}^{(i)}, \mathcal{F}_{3,\theta}^{(i)} : i = 1, \ldots, d\}$ has the following properties:

(a). There is a positive constant $C > 0$, such that for $\forall \theta \in \theta; \psi \in \Psi$,

$$\left|\frac{\partial}{\partial\theta^{(i)^2}}\mathbb{E}_\theta[s_i(z,\theta;\psi)]\right| \leqslant C < \infty.$$

(b). (*Enveloping function*) For $\forall \theta \in \Theta$, there is a function $H_\theta(z)$, such that $\mathbb{E}_\theta[H_\theta(z)^2] < \infty$, and for any function $f \in \mathcal{F}_{1,\theta} \bigcup (\cup_{i=1}^d \mathcal{F}_{2,\theta}^{(i)}) \bigcup (\cup_{i=1}^d \mathcal{F}_{3,\theta}^{(i)})$, we have

$$|f(z)| \leqslant H_\theta(z)$$

(c). (*$\theta$-uniform Donsker*) Let $\mathcal{N}(\varepsilon, \mathcal{F}, \|\cdot\|)$ denote the covering number, that is the minimal number of $\|\cdot\|$-balls of radius $\varepsilon$ needed to cover the set $\mathcal{F}$, there exists $\rho(\varepsilon) > 0$, such that

$$\int_0^1 \sqrt{\rho(\varepsilon)}d\varepsilon < \infty,$$

and

$$\sup_{\theta \in \Theta} \sup_Q \log \mathcal{N}\big(\varepsilon\|H_\theta(z)\|_{Q,2}, \mathcal{F}_\theta, L_2(Q)\big) < \rho(\varepsilon),$$

where $\mathcal{F}_\theta \in \{\mathcal{F}_{1,\theta}, \mathcal{F}_{2,\theta}^{(i)}, \mathcal{F}_{3,\theta}^{(i)} : i = 1, \ldots, d\}$ and $Q$ is any distribution on $\mathcal{Z}$.

(d). (*Vanishing optimization error*) There exists $a_n = o(1)$, such that

$$\widehat{J}_{n,k}(\theta; \widehat{\psi}(\theta)) \leqslant \min_{\psi \in \Psi} \widehat{J}_{n,k}(\theta; \psi) + a_n.$$

(e). (*Richness of class*) There exists $\psi^*(\theta)$ for each $\theta$, s.t.

$$s(z, \theta; \psi^*(\theta)) = \nabla_\theta \log p(z, \theta).$$

**Remark A.10.** *Assumption A.9(c) holds with $\rho(\varepsilon) = C \log \frac{1}{\varepsilon}$, when $\mathcal{F}_\theta$ is a VC-subgraph class for all $\mathcal{F}_\theta \in \{\mathcal{F}_{1,\theta}, \mathcal{F}_{2,\theta}^{(i)}, \mathcal{F}_{3,\theta}^{(i)} : i = 1, \ldots, d\}$.*

**Assumption A.11.** Assume that the following conditions hold:

(a). (*Smoothness*) $H_{\tilde{\theta}}(\theta)$ is $L$-joint smooth in $(\tilde{\theta}, \theta)$ for some $L < \infty$.

(b). There exists $C > 0$, such that

$$\sup_{\tilde{\theta}} \mathbb{E}_{\tilde{\theta}}\big[\nabla_\theta \ell(z; \theta_{PS})\big] \leqslant C < \infty$$

**Lemma A.12.** *Under Assumption A.9(a), we have*

$$\sup_{\theta;\psi} |J(\theta; \psi) - J_n(\theta; \psi)| \leqslant 2dC\eta.$$

*Proof.* By mean-value theorem, there exists $\eta^{(i)} \in [0, \eta]$, $1 \leqslant i \leqslant d$, such that

$$J(\theta;\psi) - J_n(\theta;\psi) = 2\sum_{i=1}^d \frac{1}{\eta}\left(\mathbb{E}_{\theta+\eta e^{(i)}}\big[s_i(z, \theta + \eta e^{(i)}; \psi)\big] - \mathbb{E}_\theta\big[s_i(z, \theta; \psi)\big]\right) - 2\sum_{i=1}^d \big[\frac{\partial}{\partial\theta^{(i)}}\mathbb{E}_\theta[s_i(z, \theta; \psi)]\big]$$

$$= 2\sum_{i=1}^d \big[\frac{\partial}{\partial\theta^{(i)}}\mathbb{E}_{\theta+\eta^{(i)}e^{(i)}}[s_i(z, \theta + \eta^{(i)}e^{(i)}; \psi)]\big] - 2\sum_{i=1}^d \big[\frac{\partial}{\partial\theta^{(i)}}\mathbb{E}_\theta[s_i(z, \theta; \psi)]\big]$$

$$\leqslant 2C\big(\sum_{i=1}^d \eta^{(i)}\big)$$

$$\leqslant 2dC\eta,$$

the first inequality follows from a direct application of mean-value theorem. ∎

**Theorem A.13.** *Let* $\Theta_{d,t} = \{\widehat{\theta}_j, \widehat{\theta}_j + \eta e^{(i)} : i = 1, \ldots, d, \ j = 1, \ldots, T\}$. *Under Assumption A.9,* *for* $\forall \theta \in \Theta_{d,t}$, *the following inequality holds*

$$\mathbb{E}_\theta \big[\|\nabla \log p(z, \theta) - s(z, \theta; \psi(\theta))\|^2 \big| \widehat{\psi}(\theta)\big] = O_p(\frac{1}{\sqrt{n}} + \frac{1}{\eta \sqrt{\min(n, k)}} + \eta + a_n).$$

*Proof.* Fix $\theta \in \Theta$, by Dudley's uniform entropy bound, c.f. Corollary 19.35 in [31], we have

$$\mathbb{E}_\theta \left[ \sup_\psi \left| \widehat{E}_{\theta,n} \big[\|s(z, \theta; \psi)\|^2\big] - E_\theta \big[\|s(z, \theta; \psi)\|^2\big] \right| \right] \lesssim \frac{1}{\sqrt{n}},$$

$$\mathbb{E}_\theta \left[ \sup_\psi \left| \widehat{E}_{\theta,n} \big[\frac{\partial}{\partial \theta^{(i)}} s_i(z, \theta; \psi)\big] - E_\theta \big[\frac{\partial}{\partial \theta^{(i)}} s_i(z, \theta; \psi)\big] \right| \right] \lesssim \frac{1}{\sqrt{n}},$$

$$\mathbb{E}_\theta \left[ \sup_\psi \left| \widehat{E}_{\theta,k} \big[s_i(z, \theta; \psi)\|^2\big] - E_\theta \big[\|s_i(z, \theta; \psi)\|^2\big] \right| \right] \lesssim \frac{1}{\sqrt{k}}.$$

Since $d, T = O(1)$, from the above results, the inequality below holds

$$\sup_{\psi \in \Psi} \sup_{\theta \in \Theta_{d,T}} \big|\widehat{J}_{n,k}(\theta; \psi) - J_n(\theta; \psi)\big| = O_p(\frac{1}{\sqrt{n}} + \frac{1}{\eta \sqrt{\min(n, k)}}),$$

by Lemma A.12, we know

$$\sup_{\psi \in \Psi} \sup_{\theta \in \Theta_{d,T}} \big|\widehat{J}_{n,k}(\theta; \psi) - J(\theta; \psi)\big| = O_p(\frac{1}{\sqrt{n}} + \frac{1}{\eta \sqrt{\min(n, k)}} + \eta). \tag{4}$$

From Assumption A.9(d), we know that for any $\theta \in \Theta_{d,T}$,

$$\widehat{J}_{n,k}(\theta; \widehat{\psi}(\theta)) \leqslant \min_{\psi \in \Psi} \widehat{J}_{n,k}(\theta; \psi) + a_n,$$

Using Assumption A.9(e), we know that $\min_{\psi \in \Psi} J(\theta; \psi) = 0$, and assume there exists $\psi^*(\theta) \in \arg\min_{\psi \in \Psi} J(\theta; \psi)$.

Take $\psi = \psi^*(\theta)$ and $\widehat{\psi}(\theta)$ respectively in inequality (4), we have

$$\widehat{J}_{n,k}(\theta; \psi^*(\theta)) = O_p(\frac{1}{\sqrt{n}} + \frac{1}{\eta \sqrt{\min(n, k)}} + \eta),$$

$$|J(\theta; \widehat{\psi}(\theta)) - \widehat{J}_{n,k}(\theta; \widehat{\psi}(\theta))| = O_p(\frac{1}{\sqrt{n}} + \frac{1}{\eta \sqrt{\min(n, k)}} + \eta).$$

Since $\widehat{J}_{n,k}(\theta; \widehat{\psi}(\theta)) \leqslant \widehat{J}_{n,k}(\theta; \psi^*(\theta)) + a_n$, we have

$$J(\theta; \widehat{\psi}(\theta)) = J(\theta; \widehat{\psi}(\theta)) - \widehat{J}_{n,k}(\theta; \widehat{\psi}(\theta)) + \widehat{J}_{n,k}(\theta; \widehat{\psi}(\theta)) - \widehat{J}_{n,k}(\theta; \psi^*(\theta)) + \widehat{J}_{n,k}(\theta; \psi^*(\theta))$$

$$= O_p(\frac{1}{\sqrt{n}} + \frac{1}{\eta \sqrt{\min(n, k)}} + \eta + a_n).$$

By definition of $J(\theta; \psi)$, we have proved

$$\mathbb{E}_\theta \big[\|\nabla \log p(z, \theta) - s(z, \theta; \psi(\theta))\|^2 \big| \widehat{\psi}(\theta)\big] = O_p(\frac{1}{\sqrt{n}} + \frac{1}{\eta \sqrt{\min(n, k)}} + \eta + a_n).$$

$\blacksquare$

Recall that
$$\nabla G(\theta_k) = -H_{\theta_k}(\theta_{k+1})^{-1} \mathbb{E}_{\theta_k} [\nabla_\theta \ell(z; \theta_{k+1}) \nabla_\theta \log p(z, \theta_k)^\top],$$
and the estimator is defined by

$$\widehat{g}_k := -H_{\widehat{\theta}_k}(\widehat{\theta}_{k+1})^{-1} \widehat{\mathbb{E}}_{\widehat{\theta}_k,n} \big[\nabla_\theta \ell(z; \widehat{\theta}_{k+1}) s(z, \widehat{\theta}_k; \widehat{\psi}(\widehat{\theta}_k))^\top\big].$$

**Theorem A.14** (Restatement of Theorem 3.6). *Under Assumption 3.1 A.1, A.9 and A.11, we have*

$$\|\widehat{g}_k - \nabla G(\theta_k)\|^2 = O_p(\frac{1}{\sqrt{n}} + \frac{1}{\eta\sqrt{\min(n,k)}} + \eta + a_n).$$

*Proof.* By Assumption 3.1,

$$\|\nabla_\theta \ell(z; \theta_{k+1}) - \nabla_\theta \ell(z; \widehat{\theta}_{k+1})\| \leqslant \beta \|\theta_{k+1} - \widehat{\theta}_{k+1}\|,$$

by Assumption 3.1, there exists $C_1 > 0$, such that

$$\|H_{\widehat{\theta}_k}(\widehat{\theta}_{k+1})^{-1}\| \leqslant C_1 < \infty.$$

The proof falls into five parts.

Let

$$\widehat{g}_{k,1} := -H_{\widehat{\theta}_k}(\widehat{\theta}_{k+1})^{-1}\widehat{\mathbb{E}}_{\widehat{\theta}_{k,n}}\left[\nabla_\theta \ell(z; \theta_{k+1})s(z, \widehat{\theta}_k; \widehat{\psi}(\widehat{\theta}_k))^\top\right].$$

**Step 1: Convergence of $\|\widehat{g}_k - \widehat{g}_{k,1}\|$.**

We have

$$\|\widehat{g}_k - \widehat{g}_{k,1}\| \lesssim \|\widehat{\theta}_{k+1} - \theta_{k+1}\|\widehat{\mathbb{E}}_{\widehat{\theta}_{k,n}}\left[\|s(z, \widehat{\theta}_k; \widehat{\psi}(\widehat{\theta}_k))\|\right]$$

$$= \frac{1}{n}\|\widehat{\theta}_{k+1} - \theta_{k+1}\| \sum_{l=1}^{n} \|s(z_{l,k}, \widehat{\theta}_k; \widehat{\psi}(\widehat{\theta}_k))\|.$$

From Assumption A.9(b), there exists $C_2 > 0$, such that

$$\mathbb{E}_{\widehat{\theta}_k}\left[\|s(z, \widehat{\theta}_k; \psi(\widehat{\theta}_k))\|^2\right] \leqslant C_2 < \infty,$$

by the law of large numbers, we know

$$\|\widehat{g}_k - \widehat{g}_{k,1}\| = O_p(\|\widehat{\theta}_{k+1} - \theta_{k+1}\|),$$

thus combine the above result with Theorem 3.4, we have

$$\|\widehat{g}_k - \widehat{g}_{k,1}\| = O_p(\frac{1}{\sqrt{n}}).$$

Let

$$\widehat{g}_{k,2} := -H_{\widehat{\theta}_k}(\widehat{\theta}_{k+1})^{-1}\mathbb{E}_{\widehat{\theta}_k}\left[\nabla_\theta \ell(z; \theta_{k+1})s(z, \widehat{\theta}_k; \widehat{\psi}(\widehat{\theta}_k))^\top\right].$$

**Step 2: Convergence of $\|\widehat{g}_{k,1} - \widehat{g}_{k,2}\|$.**

By Cauchy-Schwarz inequality, we know

$$\mathbb{E}_{\widehat{\theta}_k}\left[\left\|\nabla_\theta \ell(z; \theta_{k+1})s(z, \widehat{\theta}_k; \widehat{\psi}(\widehat{\theta}_k))^\top\right\|^2\right] \leqslant \mathbb{E}_{\widehat{\theta}_k}\left[\|\nabla_\theta \ell(z; \theta_{k+1})\|^2\right]\mathbb{E}_{\widehat{\theta}_k}\left[\|s(z, \widehat{\theta}_k; \widehat{\psi}(\widehat{\theta}_k))\|^2\right].$$

From Assumption A.9(b), we have

$$\mathbb{E}_{\widehat{\theta}_k}\left[\|s(z, \widehat{\theta}_k; \widehat{\psi}(\widehat{\theta}_k))\|^2\right] \leqslant \mathbb{E}_{\widehat{\theta}_k}[H_{\widehat{\theta}_k}^2(z)] \leqslant C < \infty,$$

where $H_{\widehat{\theta}_k}(z)$ is the enveloping function for $\widehat{\theta}_k$.

By Assumption 3.1,

$$\|\nabla_\theta \ell(z; \theta_{k+1}) - \nabla_\theta \ell(z; \theta_{PS})\| \leqslant \beta \|\theta_{k+1} - \theta_{PS}\|,$$

since

$$\|\theta_{k+1} - \theta_{PS}\| \leqslant (\frac{\varepsilon\beta}{\gamma})^{k+1}\|\theta_0 - \theta_{PS}\| \leqslant \|\theta_0 - \theta_{PS}\|,$$

thus by Assumption A.11(b),

$$\mathbb{E}_{\widehat{\theta}_k}\left[\|\nabla_\theta \ell(z; \theta_{k+1})\|^2\right] \lesssim \mathbb{E}_{\widehat{\theta}_k}\left[\|\nabla_\theta \ell(z; \theta_{PS})\|^2\right] + \|\theta_0 - \theta_{PS}\|^2 = O(1). \tag{5}$$

Hence we have

$$\mathbb{E}_{\widehat{\theta}_k}\left[\left\|\nabla_\theta \ell(z; \theta_{k+1})s(z, \widehat{\theta}_k; \widehat{\psi}(\widehat{\theta}_k))^\top\right\|^2\right] = O(1), \tag{6}$$

by Chebyshev inequality and Assumption 3.1, this lead to the following bound of $\|\widehat{g}_{k,1} - \widehat{g}_{k,2}\|$,

$$\|\widehat{g}_{k,1} - \widehat{g}_{k,2}\| \lesssim \left\| \widehat{\mathbb{E}}_{\widehat{\theta}_k, n}\left[\nabla_\theta \ell(z; \theta_{k+1}) s(z, \widehat{\theta}_k; \widehat{\psi}(\widehat{\theta}_k))^\top\right] - \mathbb{E}_{\widehat{\theta}_k}\left[\nabla_\theta \ell(z; \theta_{k+1}) s(z, \widehat{\theta}_k; \widehat{\psi}(\widehat{\theta}_k))^\top\right] \right\|$$

$$= O_p(\frac{1}{\sqrt{n}}).$$

Let

$$\widehat{g}_{k,3} := -H_{\theta_k}(\theta_{k+1})^{-1} \mathbb{E}_{\widehat{\theta}_k}\left[\nabla_\theta \ell(z; \theta_{k+1}) s(z, \widehat{\theta}_k; \widehat{\psi}(\widehat{\theta}_k))^\top\right].$$

**Step 3: Convergence of $\|\widehat{g}_{k,2} - \widehat{g}_{k,3}\|$.** By Assumption A.11(a) and (6), we have

$$\|\widehat{g}_{k,2} - \widehat{g}_{k,3}\| \lesssim \|H_{\theta_k}(\theta_{k+1})^{-1} - H_{\widehat{\theta}_k}(\widehat{\theta}_{k+1})^{-1}\|$$

$$= \left\|H_{\theta_k}(\theta_{k+1})^{-1}\big(H_{\theta_k}(\theta_{k+1}) - H_{\widehat{\theta}_k}(\widehat{\theta}_{k+1})\big) H_{\widehat{\theta}_k}(\widehat{\theta}_{k+1})^{-1}\right\|,$$

further, using Assumption 3.1 and A.11(a), we know

$$\|\widehat{g}_{k,2} - \widehat{g}_{k,3}\| \lesssim \|\widehat{\theta}_k - \theta_k\| + \|\widehat{\theta}_{k+1} - \theta_{k+1}\|,$$

thus by Theorem 3.4,

$$\|\widehat{g}_{k,2} - \widehat{g}_{k,3}\| = O_p(\frac{1}{\sqrt{n}}).$$

Let

$$\widehat{g}_{k,4} := -H_{\theta_k}(\theta_{k+1})^{-1} \mathbb{E}_{\widehat{\theta}_k}\left[\nabla_\theta \ell(z; \theta_{k+1}) \nabla_\theta \log p(z, \widehat{\theta}_k)\right].$$

**Step 4: Convergence of $\|\widehat{g}_{k,3} - \widehat{g}_{k,4}\|$.**

By Assumption 3.1 and Assumption A.1, we have

$$\|\widehat{g}_{k,3} - \widehat{g}_{k,4}\|^2 \lesssim \left\{\mathbb{E}_{\widehat{\theta}_k}\left[\|\nabla_\theta \ell(z; \theta_{k+1})\| \times \|s(z, \widehat{\theta}_k; \widehat{\psi}(\widehat{\theta}_k)) - \nabla_\theta \log p(z, \widehat{\theta}_k)\|\right]\right\}^2$$

$$\leqslant \mathbb{E}_{\widehat{\theta}_k}\left[\|\nabla_\theta \ell(z; \theta_{k+1})\|^2\right]$$

$$\times \mathbb{E}_{\widehat{\theta}_k}\left[\|s(z, \widehat{\theta}_k; \widehat{\psi}(\widehat{\theta}_k)) - \nabla_\theta \log p(z, \widehat{\theta}_k)\|^2\right] \quad \text{(by Cauchy} - \text{Schwarz inequality)}$$

$$\lesssim \mathbb{E}_{\widehat{\theta}_k}\left[\|s(z, \widehat{\theta}_k; \widehat{\psi}(\widehat{\theta}_k)) - \nabla_\theta \log p(z, \widehat{\theta}_k)\|^2\right] \quad \text{(by formula (5))}$$

$$= O_p(\frac{1}{\sqrt{n}} + \frac{1}{\eta\sqrt{\min(n, k)}} + \eta + a_n) \quad \text{(by Theorem A.13).}$$

**Step 5: Convergence of $\|\widehat{g}_k - \nabla G(\theta_k)\|$.**

Finally, recall that

$$\nabla G(\theta_k) = -H_{\theta_k}(\theta_{k+1})^{-1} \mathbb{E}_{\widehat{\theta}_k}\left[\nabla_\theta \ell(z; \theta_{k+1}) \nabla_\theta \log p(z, \theta_k)\right],$$

by Assumption 3.1 and A.1,

$$\|\widehat{g}_{k,4} - \nabla G(\theta_k)\| \lesssim \mathbb{E}_{\widehat{\theta}_k}\left[\|\nabla_\theta \ell(z; \theta_{k+1})\|\right] \times \|\widehat{\theta}_k - \theta_k\|$$

$$\lesssim \|\widehat{\theta}_k - \theta_k\|$$

$$= O_p(\frac{1}{\sqrt{n}}) \quad \text{(By Theorem 3.4).}$$

Combining the above results, we thus have proved

$$\|\widehat{g}_k - \nabla G(\theta_k)\|^2 = O_p(\frac{1}{\sqrt{n}} + \frac{1}{\eta\sqrt{\min(n, k)}} + \eta + a_n).$$

■

## A.4 Details of Section 4: Theory of Prediction-Powered Inference under Performativity

Without loss of generality, we let $N_t = N$ and $n_t = n$ for all $t \in [T]$.

Let us denote

$$\mathcal{L}_{\tilde{\theta}}(\theta) := \mathbb{E}_{(x,y)\sim\mathcal{D}(\tilde{\theta})}\ell(x,y;\theta), \quad \mathcal{L}_{\tilde{\theta}}^{f,\lambda}(\theta) := \mathcal{L}_{\tilde{\theta},n}(\theta) + \lambda \cdot (\widetilde{\mathcal{L}}_{\tilde{\theta},N}^{f}(\theta) - \mathcal{L}_{\tilde{\theta},n}^{f}(\theta)),$$

where

$$\mathcal{L}_{\tilde{\theta},n}(\theta) := \frac{1}{n}\sum_{i=1}^{n}\ell(x_i,y_i;\theta), \; \mathcal{L}_{\tilde{\theta},n}^{f}(\theta) := \frac{1}{n}\sum_{i=1}^{n}\ell\left((x_i,f(x_i));\theta\right), \; \widetilde{\mathcal{L}}_{\tilde{\theta},N}^{f}(\theta) := \frac{1}{N}\sum_{i=1}^{N}\ell\left((x_i^u,f(x_i^u));\theta\right).$$

Here the samples $(x_i, y_i) \sim \mathcal{D}(\tilde{\theta})$ and $x_i^u \sim \mathcal{D}_{\mathcal{X}}(\tilde{\theta})$ are drawn from the distribution under $\tilde{\theta}$. Recall that we have defined $\Sigma_{\lambda,\tilde{\theta}}(\theta) = H_{\tilde{\theta}}(\theta)^{-1}\left(rV_{\lambda,\tilde{\theta}}^{f}(\theta) + V_{\lambda,\tilde{\theta}}(\theta)\right)H_{\tilde{\theta}}(\theta)^{-1}$ before Theorem 4.1 (in the following we sometimes omit $r$ for simplicity).

**Theorem A.15** (Consistency of $\widehat{\theta}_t^{\text{PPI}}$). *Under Assumption 3.1 and A.4, if $\varepsilon < \frac{\gamma}{\beta}$, then for any given $T \geqslant 0$, we have that for all $t \in [T]$,*

$$\widehat{\theta}_{t+1}^{PPI}(\lambda_t) \xrightarrow{P} \theta_{t+1}.$$

*Proof.* Let us denote $\widehat{G}_{\lambda}^{f}(\theta) := \arg\min_{\theta'\in\Theta} \frac{\lambda}{N}\sum_{i=1}^{N}\ell(x_i^u,f(x_i^u);\theta') + \frac{1}{n}\sum_{i=1}^{n}\left(\ell(x_i,y_i;\theta') - \lambda\ell(x_i,f(x_i);\theta')\right)$, where the samples $(x_i,y_i) \sim \mathcal{D}(\theta)$ and $x_i^u \sim \mathcal{D}_{\mathcal{X}}(\theta)$ are drawn for some parameter $\theta$ along the dynamic trajectory $\theta_0 \to \widehat{\theta}_1 \to \cdots \widehat{\theta}_t \to \cdots$.

$$\begin{aligned}
\|\theta_t - \widehat{\theta}_t^{\text{PPI}}\| &= \|G(\theta_{t-1}) - \widehat{G}_{\lambda_t}^{f}(\widehat{\theta}_{t-1}^{\text{PPI}})\| \\
&\leqslant \|G(\widehat{\theta}_{t-1}^{\text{PPI}}) - \widehat{G}_{\lambda_t}^{f}(\widehat{\theta}_{t-1}^{\text{PPI}})\| + \|G(\theta_{t-1}) - G(\widehat{\theta}_{t-1}^{\text{PPI}})\| \\
&\leqslant \|G(\widehat{\theta}_{t-1}^{\text{PPI}}) - \widehat{G}_{\lambda_t}^{f}(\widehat{\theta}_{t-1}^{\text{PPI}})\| + \varepsilon\frac{\beta}{\gamma}\|\theta_{t-1} - \widehat{\theta}_{t-1}^{\text{PPI}}\|,
\end{aligned}$$

where the last inequality follows from the results derived by [25], under Assumption 3.1, we have $\|G(\theta) - G(\theta')\| \leqslant \frac{\varepsilon\beta}{\gamma}\|\theta - \theta'\|$.

Notice that $\mathbb{E}(\mathcal{L}_{\widehat{\theta}_{t-1}^{\text{PPI}}}^{f,\lambda_t}(\theta)) = \mathcal{L}_{\widehat{\theta}_{t-1}^{\text{PPI}}}(\theta)$. By local Lipschitz condition, there exists $\varepsilon_0 > 0$ such that

$$\sup_{\theta:\|\theta-G(\widehat{\theta}_{t-1}^{\text{PPI}})\|\leqslant\varepsilon_0}|\mathcal{L}_{\widehat{\theta}_{t-1}^{\text{PPI}}}^{f,\lambda_t}(\theta) - \mathcal{L}_{\widehat{\theta}_{t-1}^{\text{PPI}}}(\theta)| \xrightarrow{P} 0.$$

Since $\ell$ is strongly convex for any $\theta$, $G(\widehat{\theta}_{t-1}^{\text{PPI}})$ is unique. Then we know that there exists $\delta$ such that $\mathcal{L}_{\widehat{\theta}_{t-1}^{\text{PPI}}}^{f,\lambda_t}(\theta) - \mathcal{L}_{\widehat{\theta}_{t-1}^{\text{PPI}}}(G(\widehat{\theta}_{t-1}^{\text{PPI}})) > \delta$ for all $\theta$ in $\{\theta \mid \|\theta - G(\widehat{\theta}_{t-1}^{\text{PPI}})\| = \varepsilon_0\}$. Then it follows that:

$$\begin{aligned}
&\inf_{\|\theta-G(\widehat{\theta}_{t-1}^{\text{PPI}})\|=\varepsilon_0} \mathcal{L}_{\widehat{\theta}_{t-1}^{\text{PPI}}}^{f,\lambda_t}(\theta) - \mathcal{L}_{\widehat{\theta}_{t-1}^{\text{PPI}}}^{f,\lambda_t}(G(\widehat{\theta}_{t-1}^{\text{PPI}})) \\
&= \inf_{\|\theta-G(\widehat{\theta}_{t-1}^{\text{PPI}})\|=\varepsilon_0} \left((\mathcal{L}_{\widehat{\theta}_{t-1}^{\text{PPI}}}^{f,\lambda_t}(\theta) - \mathcal{L}_{\widehat{\theta}_{t-1}^{\text{PPI}}}(\theta)) + (\mathcal{L}_{\widehat{\theta}_{t-1}^{\text{PPI}}}(\theta) - \mathcal{L}_{\widehat{\theta}_{t-1}^{\text{PPI}}}(G(\widehat{\theta}_{t-1}^{\text{PPI}}))) \right. \\
&\quad + \left. (\mathcal{L}_{\widehat{\theta}_{t-1}^{\text{PPI}}}(G(\widehat{\theta}_{t-1}^{\text{PPI}})) - \mathcal{L}_{\widehat{\theta}_{t-1}^{\text{PPI}}}^{f,\lambda_t}(G(\widehat{\theta}_{t-1}^{\text{PPI}})))\right) \\
&\geqslant \delta - o_P(1).
\end{aligned}$$

Then we consider any fixed $\theta$ such that $\|\theta - G(\widehat{\theta}_{t-1}^{\text{PPI}})\| \geqslant \varepsilon_0$ it follows that

$$\begin{aligned}
\mathcal{L}_{\widehat{\theta}_{t-1}^{\text{PPI}}}^{f,\lambda_t}(\theta) - \mathcal{L}_{\widehat{\theta}_{t-1}^{\text{PPI}}}^{f,\lambda_t}(G(\widehat{\theta}_{t-1}^{\text{PPI}})) &\geqslant \frac{\theta - G(\widehat{\theta}_{t-1}^{\text{PPI}})}{\omega - G(\widehat{\theta}_{t-1}^{\text{PPI}})}\left(\mathcal{L}_{\widehat{\theta}_{t-1}^{\text{PPI}}}^{f,\lambda_t}(\omega) - \mathcal{L}_{\widehat{\theta}_{t-1}^{\text{PPI}}}^{f,\lambda_t}(G(\widehat{\theta}_{t-1}^{\text{PPI}}))\right) \\
&\geqslant \frac{\|\theta - G(\widehat{\theta}_{t-1}^{\text{PPI}})\|}{\varepsilon_0}(\delta - o_P(1)) \geqslant \delta - o_P(1),
\end{aligned}$$

where the first inequality holds for any $\omega$ by the convexity condition of $\mathcal{L}^{f,\lambda_t}_{\widehat{\theta}^{\mathrm{PPI}}_{t-1}}(\theta)$, and the second inequality holds as we take $\omega = \frac{\theta - G(\widehat{\theta}^{\mathrm{PPI}}_{t-1})}{\|\theta - G(\widehat{\theta}^{\mathrm{PPI}}_{t-1})\|}\varepsilon_0 + G(\widehat{\theta}^{\mathrm{PPI}}_{t-1})$ and using the above result. Thus no $\theta$ such that $\|\theta - G(\widehat{\theta}^{\mathrm{PPI}}_{t-1})\| = \varepsilon_0$ can be the minimizer of $\mathcal{L}^{f,\lambda_t}_{\widehat{\theta}^{\mathrm{PPI}}_{t-1}}(\theta)$. Then $\|G(\widehat{\theta}^{\mathrm{PPI}}_{t-1}) - \widehat{G}^f_{\lambda_t}(\widehat{\theta}^{\mathrm{PPI}}_{t-1})\| \xrightarrow{P} 0$.

We then have, for a given $T \geqslant 0$, we have that for all $t \in [T]$,

$$\|\widehat{\theta}^{\mathrm{PPI}}_t - \theta_t\| \leqslant \sum_{i=0}^{t}(\varepsilon\frac{\beta}{\gamma})^{t-i}\|G(\widehat{\theta}^{\mathrm{PPI}}_i) - \widehat{G}^f_{\lambda_i}(\widehat{\theta}^{\mathrm{PPI}}_i)\| \xrightarrow{P} 0.$$

Thus, we conclude that $\widehat{\theta}^{\mathrm{PPI}}_t \xrightarrow{P} \theta_t$. ∎

**Theorem A.16** (Central Limit Theorem of $\widehat{\theta}^{\mathrm{PPI}}_t(\lambda_t)$, Restatement of Theorem 4.1)**.** *Under Assumption 3.1, A.4, and A.5, if $\varepsilon < \frac{\gamma}{\beta}$ and $\frac{n}{N} \to r$ for some $r \geqslant 0$, then for any given $T \geqslant 0$, we have that for all $t \in [T]$,*

$$\sqrt{n}\big(\widehat{\theta}^{PPI}_t(\lambda_t) - \theta_t\big) \xrightarrow{D} \mathcal{N}\Big(0, V^{PPI}_t(\{\lambda_j, \theta_j\}^t_{j=1}; r)\Big)$$

*with*

$$V^{PPI}_t(\{\lambda_j, \theta_j\}^t_{j=1}; r) = \sum_{i=1}^{t}\left[\prod_{k=i}^{t-1}\nabla G(\theta_k)\right]\Sigma_{\lambda_i,\theta_{i-1}}(\theta_i; r)\left[\prod_{k=i}^{t-1}\nabla G(\theta_k)\right]^\top.$$

*Proof.* Let us denote the variance terms by $V^{\mathrm{PPI}}_t$ for simplicity, while omitting explicit dependence on parameters in the notation. Let $U_t := \sqrt{n}(\widehat{\theta}^{\mathrm{PPI}}_t - \theta_t)$ and denote $\tilde{\theta}_t = G(\widehat{\theta}^{\mathrm{PPI}}_{t-1})$. We make the following decomposition:

$$\widehat{\theta}^{\mathrm{PPI}}_t - \theta_t = \underbrace{(\tilde{\theta}_t - \theta_t)}_{(1)} + \underbrace{(\widehat{\theta}^{\mathrm{PPI}}_t - \tilde{\theta}_t)}_{(2)}.$$

**Step 1: Conditional distribution of $U_t|U_{t-1}$.**

For term (1), we have

$$\sqrt{n}(\tilde{\theta}_t - \theta_t) = \sqrt{n}(G(\widehat{\theta}^{\mathrm{PPI}}_{t-1}) - G(\theta_{t-1})).$$

For term (2), the empirical process analysis in [1] establishes that

$$\sqrt{n}(\widehat{\theta}^{\mathrm{PPI}}_t - \tilde{\theta}_t)|\widehat{\theta}^{\mathrm{PPI}}_{t-1} \xrightarrow{D} \mathcal{N}(0, \Sigma_{\lambda_t, \widehat{\theta}^{\mathrm{PPI}}_{t-1}}(\tilde{\theta}_t; r)),$$

where the variance is given by

$$\Sigma_{\widehat{\theta}^{\mathrm{PPI}}_{t-1}}(\tilde{\theta}_t; r) = H_{\widehat{\theta}^{\mathrm{PPI}}_{t-1}}(\tilde{\theta}_t)^{-1}\left(rV^f_{\lambda_t, \widehat{\theta}^{\mathrm{PPI}}_{t-1}}(\tilde{\theta}_t) + V_{\lambda_t, \widehat{\theta}^{\mathrm{PPI}}_{t-1}}(\tilde{\theta}_t)\right)H_{\widehat{\theta}^{\mathrm{PPI}}_{t-1}}(\tilde{\theta}_t)^{-1}.$$

Conditioning on $\widehat{\theta}^{\mathrm{PPI}}_{t-1}$, for any function $h$, we use the following shorthand notations:

$$\mathbb{E}_n h := \frac{1}{n}\sum_{i=1}^{n}h(x_i, y_i), \quad \mathbb{G}_n h := \sqrt{n}(\mathbb{E}_n h - \mathbb{E}_{(x,y)\sim\mathcal{D}(\widehat{\theta}^{\mathrm{PPI}}_{t-1})}[h(x,y)]),$$

$$\widehat{\mathbb{E}}^f_N h := \frac{1}{N}\sum_{i=1}^{N}h(x^u_i, f(x^u_i)), \quad \widehat{\mathbb{G}}^f_N h := \sqrt{N}(\widehat{\mathbb{E}}_N h - \mathbb{E}_{x\sim\mathcal{D}_\mathcal{X}(\widehat{\theta}^{\mathrm{PPI}}_{t-1})}[h(x, f(x))]),$$

$$\widehat{\mathbb{E}}^f_n h := \frac{1}{n}\sum_{i=1}^{n}h(x_i, f(x_i)), \quad \widehat{\mathbb{G}}^f_n h := \sqrt{n}(\widehat{\mathbb{E}}_n h - \mathbb{E}_{x\sim\mathcal{D}_\mathcal{X}(\widehat{\theta}^{\mathrm{PPI}}_{t-1})}[h(x, f(x))]).$$

Note that $\tilde{\theta}_t = G(\widehat{\theta}^{\mathrm{PPI}}_{t-1})$. Recall that

$$\mathcal{L}_{\tilde{\theta}}(\theta) := \mathbb{E}_{(x,y)\sim\mathcal{D}(\tilde{\theta})}\ell(x, y; \theta), \quad \mathcal{L}^{f,\lambda}_{\tilde{\theta}}(\theta) := \mathcal{L}_{\tilde{\theta},n}(\theta) + \lambda\cdot(\widetilde{\mathcal{L}}^f_{\tilde{\theta},N}(\theta) - \mathcal{L}^f_{\tilde{\theta},n}(\theta)).$$

Under the assumptions, Lemma 19.31 in [31] implies that for every sequence $h_n = O_P(1)$, we have

$$\mathbb{G}_n \left[ \sqrt{n} \left( \ell(x, y; \tilde{\theta}_t + \frac{h_n}{\sqrt{n}}) - \ell(x, y; \tilde{\theta}_t) \right) - h_n^\top \nabla_\theta \ell(x, y; \tilde{\theta}_t) \right] \xrightarrow{P} 0,$$

$$\widehat{\mathbb{G}}_N^f \left[ \sqrt{n} \left( \ell(x, y; \tilde{\theta}_t + \frac{h_n}{\sqrt{n}}) - \ell(x, y; \tilde{\theta}_t) \right) - h_n^\top \nabla_\theta \ell(x, y; \tilde{\theta}_t) \right] \xrightarrow{P} 0,$$

$$\widehat{\mathbb{G}}_n^f \left[ \sqrt{n} \left( \ell(x, y; \tilde{\theta}_t + \frac{h_n}{\sqrt{n}}) - \ell(x, y; \tilde{\theta}_t) \right) - h_n^\top \nabla_\theta \ell(x, y; \tilde{\theta}_t) \right] \xrightarrow{P} 0.$$

Applying second-order Taylor expansion, we obtain that

$$n \mathbb{E}_n \left( \ell(x, y; \tilde{\theta}_t + \frac{h_n}{\sqrt{n}}) - \ell(x, y; \tilde{\theta}_t) \right) = n \left( \mathcal{L}_{\widehat{\theta}_{t-1}^{\mathrm{PPI}}}(\tilde{\theta}_t + \frac{h_n}{\sqrt{n}}) - \mathcal{L}_{\widehat{\theta}_{t-1}^{\mathrm{PPI}}}(\tilde{\theta}_t) \right) + h_n^\top \mathbb{G}_n \nabla_\theta \ell(x, y; \tilde{\theta}_t) + o_p(1)$$

$$= \frac{1}{2} h_n^\top H_{\widehat{\theta}_{t-1}^{\mathrm{PPI}}}(\tilde{\theta}_t) h_n + h_n^\top \mathbb{G}_n \nabla_\theta \ell(x, y; \tilde{\theta}_t) + o_p(1).$$

Based on similar calculation of the previous two terms, we can obtain that:

$$n \left( \mathcal{L}_{\widehat{\theta}_{t-1}^{\mathrm{PPI}}}^{f,\lambda}(\tilde{\theta}_t + \frac{h_n}{\sqrt{n}}) - \mathcal{L}_{\widehat{\theta}_{t-1}^{\mathrm{PPI}}}^{f,\lambda}(\tilde{\theta}_t) \right)$$

$$= \frac{1}{2} h_n^\top H_{\widehat{\theta}_{t-1}^{\mathrm{PPI}}}(\tilde{\theta}_t) h_n + h_n^\top \left( \mathbb{G}_n + \lambda \sqrt{\frac{n}{N}} \widehat{\mathbb{G}}_N^f - \lambda \widehat{\mathbb{G}}_n^f \right) \nabla_\theta \ell(x, y; \tilde{\theta}_t) + o_p(1).$$

By considering $h_n^* = \sqrt{n}(\widehat{\theta}_t^{\mathrm{PPI}} - \tilde{\theta}_t)$ and $h_n = -H_{\widehat{\theta}_{t-1}^{\mathrm{PPI}}}(\tilde{\theta}_t)^{-1} \left( \mathbb{G}_n + \lambda \sqrt{\frac{n}{N}} \widehat{\mathbb{G}}_N^f - \lambda \widehat{\mathbb{G}}_n^f \right) \nabla_\theta \ell(x, y; \tilde{\theta}_t)$,
Corollary 5.53 in [31] implies they are $O_P(1)$ and we obtain that

$$n \left( \mathcal{L}_{\widehat{\theta}_{t-1}^{\mathrm{PPI}}}^{f,\lambda}(\widehat{\theta}_t^{\mathrm{PPI}}) - \mathcal{L}_{\widehat{\theta}_{t-1}^{\mathrm{PPI}}}^{f,\lambda}(\tilde{\theta}_t) \right) = \frac{1}{2} h_n^{*\top} H_{\widehat{\theta}_{t-1}^{\mathrm{PPI}}}(\tilde{\theta}_t) h_n^* + h_n^{*\top} \left( \mathbb{G}_n + \lambda \sqrt{\frac{n}{N}} \widehat{\mathbb{G}}_N^f - \lambda \widehat{\mathbb{G}}_n^f \right) \nabla_\theta \ell(x, y; \tilde{\theta}_t) + o_p(1)$$

$$n \left( \mathcal{L}_{\widehat{\theta}_{t-1}^{\mathrm{PPI}}}^{f,\lambda}(\tilde{\theta}_t + \frac{h_n}{\sqrt{n}}) - \mathcal{L}_{\widehat{\theta}_{t-1}^{\mathrm{PPI}}}^{f,\lambda}(\tilde{\theta}_t) \right) = -\frac{1}{2} h_n^\top H_{\widehat{\theta}_{t-1}^{\mathrm{PPI}}}(\tilde{\theta}_t) h_n + o_P(1).$$

Since $\widehat{\theta}_t^{\mathrm{PPI}}$ is the minimizer of $\mathcal{L}_{\widehat{\theta}_{t-1}^{\mathrm{PPI}}}^{f,\lambda}$, the first term is smaller than the second term. We can rearrange the terms and obtain:

$$\frac{1}{2}(h_n^* - h_n)^T H_{\widehat{\theta}_{t-1}^{\mathrm{PPI}}}(\tilde{\theta}_t)(h_n^* - h_n) = o_P(1),$$

which leads to $h_n^* - h_n = O_P(1)$. Then the above asymptotic normality result follows directly by applying the central limit theorem (CLT) to the following terms, conditioning on $\widehat{\theta}_{t-1}^{\mathrm{PPI}}$:

$$\sqrt{n}(\widehat{\theta}_t^{\mathrm{PPI}} - \tilde{\theta}_t)|\widehat{\theta}_{t-1}^{\mathrm{PPI}} = -H_{\widehat{\theta}_{t-1}^{\mathrm{PPI}}}(\tilde{\theta}_t)^{-1}(S_1 + S_2) + o_P(1),$$

$$S_1 = \lambda_t \sqrt{\frac{n}{N}} \sqrt{\frac{1}{N}} \sum_{i=1}^N \left( \nabla_\theta \ell(x_{t,i}^u, f(x_{t,i}^u); \tilde{\theta}_t) - \mathbb{E}_{x \sim \mathcal{D}_\mathcal{X}(\widehat{\theta}_{t-1}^{\mathrm{PPI}})} \nabla_\theta \ell(x, f(x); \tilde{\theta}_t) \right),$$

$$S_2 = \sqrt{\frac{1}{n}} \sum_{i=1}^n \left( \nabla_\theta \ell(x_{t,i}, y_{t,i}; \tilde{\theta}_t) - \lambda_t \nabla_\theta \ell(x_{t,i}, f(x_{t,i}); \tilde{\theta}_t) \right.$$

$$\left. - \mathbb{E}_{(x,y) \sim \mathcal{D}(\widehat{\theta}_{t-1}^{\mathrm{PPI}})} [\nabla_\theta \ell(x, y; \tilde{\theta}_t) - \lambda_t \nabla_\theta \ell(x, f(x); \tilde{\theta}_t)] \right).$$

Note that, conditioning on $\widehat{\theta}_{t-1}^{\mathrm{PPI}}$, (1) is a constant. Therefore, (1) and (2) follow a joint Gaussian distribution. Consequently, given $U_{t-1}$, the conditional distribution of $U_t$ is given by:

$$U_t | U_{t-1} = \sqrt{n}(\widehat{\theta}_t^{\mathrm{PPI}} - \theta_t)|\widehat{\theta}_{t-1}^{\mathrm{PPI}}$$

$$= \sqrt{n}(\tilde{\theta}_t - \theta_t) + \sqrt{n}(\widehat{\theta}_t^{\mathrm{PPI}} - \tilde{\theta}_t)|\widehat{\theta}_{t-1}^{\mathrm{PPI}}$$

$$= \sqrt{n}(G(\widehat{\theta}_{t-1}^{\mathrm{PPI}}) - G(\theta_{t-1})) + \sqrt{n}(\widehat{\theta}_t^{\mathrm{PPI}} - \tilde{\theta}_t)|\widehat{\theta}_{t-1}^{\mathrm{PPI}}$$

$$\xrightarrow{D} \mathcal{N}\left(\sqrt{n}(G(\widehat{\theta}_{t-1}^{\mathrm{PPI}}) - G(\theta_{t-1})), \Sigma_{\lambda_t, \widehat{\theta}_{t-1}^{\mathrm{PPI}}}(\tilde{\theta}_t; r)\right).$$

$$= \mathcal{N}\left(\sqrt{n}(G(\frac{U_{t-1}}{\sqrt{n}} + \theta_{t-1}) - G(\theta_{t-1})), \Sigma_{\lambda_t, \frac{U_{t-1}}{\sqrt{n}} + \theta_{t-1}}(G(\frac{U_{t-1}}{\sqrt{n}} + \theta_{t-1}); r)\right).$$

For later references, we denote $U_t \mid U_{t-1} \xrightarrow{D} \mathcal{N}(\mu(U_{t-1}), \Sigma(U_{t-1}; r))$.

**Step 2: Marginal distribution of $U_t$.** We calculate the characteristic function of $U_t$ by induction. To begin with, we directly have

$$X_1 \xrightarrow{D} \mathcal{N}(0, V_1^{\mathrm{PPI}}), \quad V_1^{\mathrm{PPI}} = \Sigma_{\lambda_0, \theta_0}(\theta_1; r).$$

Now, assume that $U_{t-1} \xrightarrow{D} \mathcal{N}(0, V_{t-1}^{\mathrm{PPI}})$, we derive the joint distribution of $(U_t, U_{t-1})$ and marginal distribution of $U_t$. Then we have, the characteristics functions $\phi$ and the probability density function $p$ of distributions $U_{t-1}$ and $U_t \mid U_{t-1}$ follow:

$$\phi_{U_{t-1}}(s) \to \phi_{\mathcal{N}(0, V_{t-1}^{\mathrm{PPI}})}(s) = \exp(-\frac{1}{2}s^T V_{t-1}^{\mathrm{PPI}} s), \quad p_{U_{t-1}}(u) = \frac{1}{(2\pi)^d} \int e^{-iz^T u} \phi_{U_{t-1}}(z) dz$$

$$\phi_{U_t \mid U_{t-1}}(s) \to \phi_{\mathcal{N}(\mu(U_{t-1}), \Sigma(U_{t-1}; r))}(s) = \exp(is^T \mu(U_{t-1}) - \frac{1}{2}s^T \Sigma(U_{t-1}; r)s).$$

Then according to the proof of vanilla CLT under performativity in Section A.2, we have:

$$\phi_{U_t}(s) = \frac{1}{(2\pi)^{\frac{d}{2}} \det |V_{t-1}^{\mathrm{PPI}}|} \int \exp\left(is^T \mu(U_{t-1}) - \frac{1}{2}s^T \Sigma(U_{t-1}; r)s - \frac{1}{2}u^T V_{t-1}^{\mathrm{PPI}} u\right) du.$$

Apply dominant convergence theorem to $\lim_{n \to \infty} \phi_{U_t}(s)$, we have:

$$\lim_{n \to \infty} \phi_{U_t}(s) = \lim_{n \to \infty} \frac{1}{(2\pi)^{\frac{d}{2}} \det |V_{t-1}^{\mathrm{PPI}}|} \int \exp\left(is^T \mu(U_{t-1}) - \frac{1}{2}s^T \Sigma(U_{t-1}; r)s - \frac{1}{2}u^T V_{t-1}^{\mathrm{PPI}} u\right) du$$

$$= \lim_{n \to \infty} \frac{1}{(2\pi)^{\frac{d}{2}} \det |V_{t-1}^{\mathrm{PPI}}|} \int \exp\Big(is^T \sqrt{n}\big(G(\tfrac{U_{t-1}}{\sqrt{n}} + \theta_{t-1}) - G(\theta_{t-1})\big)$$

$$- \tfrac{1}{2}s^T \Sigma_{\lambda_t, \frac{U_{t-1}}{\sqrt{n}} + \theta_{t-1}}\big(G(\tfrac{U_{t-1}}{\sqrt{n}} + \theta_{t-1}); r\big)s - \tfrac{1}{2}u^T V_{t-1}^{\mathrm{PPI}} u\Big) du$$

$$= \frac{1}{(2\pi)^{\frac{d}{2}} \det |V_{t-1}^{\mathrm{PPI}}|} \int \exp\Big(is^T \sqrt{n}\big(G(\tfrac{U_{t-1}}{\sqrt{n}} + \theta_{t-1}) - G(\theta_{t-1})\big)$$

$$- \tfrac{1}{2}s^T \Sigma_{\lambda_t, \frac{U_{t-1}}{\sqrt{n}} + \theta_{t-1}}\big(G(\tfrac{U_{t-1}}{\sqrt{n}} + \theta_{t-1}); r\big)s - \tfrac{1}{2}u^T V_{t-1}^{\mathrm{PPI}} u\Big) du$$

$$= \frac{1}{(2\pi)^{\frac{d}{2}} \det |V_{t-1}^{\mathrm{PPI}}|} \int \exp\Big(is^T \nabla G(\theta_{t-1})u - \tfrac{1}{2}s^T \Sigma_{\lambda_t, \theta_{t-1}}(G(\theta_{t-1}); r)s - \tfrac{1}{2}u^T V_{t-1}^{\mathrm{PPI}} u\Big) du$$

$$= \exp\Big(-\tfrac{1}{2}s^T \nabla G(\theta_{t-1}) V_{t+1}^{\mathrm{PPI}} \nabla G(\theta_{t-1})^T s - \tfrac{1}{2}s^T \Sigma_{\lambda_t, \theta_{t-1}}(\theta_t; r)s\Big),$$

which is the characteristic function of $\mathcal{N}(0, V_t^{\mathrm{PPI}})$, where $V_t^{\mathrm{PPI}} = \nabla G(\theta_{t-1}) V_{t-1}^{\mathrm{PPI}} \nabla G(\theta_{t-1})^\top + \Sigma_{\lambda_t, \theta_{t-1}}(\theta_t; r)$. Here we use the fact that $\lim_{n \to \infty} \sqrt{n}\left(G(\frac{y}{\sqrt{n}} + \theta_{t-1}) - G(\theta_{t-1})\right) = \nabla G(\theta_{t-1})y$, and the dominant convergence theorem holds as we have

$$|\exp(is^T \sqrt{n}(G(\frac{U_{t-1}}{\sqrt{n}} + \theta_{t-1}) - G(\theta_{t-1})) - \frac{1}{2}s^T \Sigma_{\lambda_t, \frac{U_{t-1}}{\sqrt{n}} + \theta_{t-1}}(G(\frac{U_{t-1}}{\sqrt{n}} + \theta_{t-1}); r)s - \frac{1}{2}u^T V_{t-1}^{\mathrm{PPI}} u)|$$

$$\leqslant |\exp(-\frac{1}{2}u^T V_{t-1}^{\mathrm{PPI}} u)|.$$

Thus we conclude by induction that

$$U_t \xrightarrow{D} \mathcal{N}\left(0, V_t^{\mathrm{PPI}}\right),$$

$$V_t^{\mathrm{PPI}} = \sum_{i=1}^{t} \left[\prod_{k=i}^{t-1} \nabla G(\theta_k)\right] \Sigma_{\lambda_{i-1}, \theta_{i-1}}(\theta_i; r) \left[\prod_{k=i}^{t-1} \nabla G(\theta_k)\right]^\top.$$

And we have:

$$\nabla G(\theta_k) = -\left[\underset{(x,y)\sim\mathcal{D}(\theta_k)}{\mathbb{E}}\nabla_\theta^2\ell(x,y;\theta_{k+1})\right]^{-1}\left(\nabla_{\tilde\theta}\underset{(x,y)\sim\mathcal{D}(\theta_k)}{\mathbb{E}}\nabla_\theta\ell(x,y;\theta_{k+1})\right)$$

$$= -H_{\theta_k}(\theta_{k+1})^{-1}\left(\nabla_{\tilde\theta}\underset{(x,y)\sim\mathcal{D}(\theta_k)}{\mathbb{E}}\nabla_\theta\ell(x,y;\theta_{k+1})\right)$$

$$= -H_{\theta_k}(\theta_{k+1})^{-1}\mathbb{E}_{z\sim\mathcal{D}(\theta_k)}[\nabla_\theta\ell(z,\theta_{k+1})\nabla_\theta\log p(z,\theta_k)^\top].$$

■

## B  Experimental Details

### B.1  Additional Experimental Details

As described in Section 5, we construct simulation studies on a performative linear regression problem, where data are sampled from $D(\theta)$ as

$$y = \alpha^\top x + \mu^\top\theta + \nu,\ x\sim\mathcal{N}(\mu_x,\Sigma_x),\ \nu\sim\mathcal{N}(0,\sigma_y^2).$$

At each time step $t$, the label $y_t$ is updated with $\widehat{\theta}_{t-1}$ via the above equation, and then $\widehat{\theta}_t$ is obtained by empirical repeated risk minimization with the updated data $z_t = (x_t, y_t)$. The objective of this task is to provide inference on an unbiased $\widehat{\theta}_t$ with low variance, that is, the ground-truth $\theta_t$ is covered by the confidence region of $\widehat{\theta}_t$ with high probability, and the width of this confidence region is small.

Given a set of labeled data, we can obtain the underlying $\theta_t$ as

$$\theta_t = (\Sigma_x + \mu_x\mu_x^\top + \gamma I_d)^{-1}\left(\mu_x\mu^\top\theta_{t-1} + (\Sigma_x + \mu_x\mu_x^\top)\alpha\right). \tag{7}$$

To compute the coverage and width of a confidence region $\widehat{\mathcal{R}}_t(n,\delta)$ for $\theta_t$, we run 1000 independent trials. For each trial $j$, we sample $\widehat{\theta}_{t,j}$ together with with its estimated variance $\widehat{V}_{t,j}$, and construct two-sided normal intervals for each coordinate $i = 1, \dots, d$:

$$\left[\widehat{\theta}_{t,j}^{(i)} \pm q_{1-\frac{\delta}{2d}}\sqrt{\widehat{V}_{t,j}^{(i)}/n}\right],\quad q_{1-\frac{\delta}{2d}} = \Phi^{-1}(1 - \frac{\delta}{2d}),$$

where $d = 2$ is the parameter dimension, $\delta = 0.1$ the significance level, $n$ the data size, and $\Phi^{-1}$ the standard normal quantile. The interval width of each trial is averaged over $d$ coordinate intervals, and we count this trial as covered if the ground-truth $\theta_t$ lies inside *all* $d$ coordinate intervals simultaneously. Finally, we report the average width and coverage rate over all trials.

Similarly, to compute the coverage for performative stable point $\theta_{\mathrm{PS}}$, we can obtain the close-form $\theta_{\mathrm{PS}}$ for this task as follows:

$$\theta_{\mathrm{PS}} = (\Sigma_x + \mu_x\mu_x^\top - \mu_x\mu^\top + \gamma I_d)^{-1}(\Sigma_x + \mu_x\mu_x^\top). \tag{8}$$

As defined in Corollary 3.7, the confidence region for $\theta_{\mathrm{PS}}$ is constructed with

$$\widehat{\mathcal{R}}_t(n,\delta) + \mathcal{B}\left(0, 2B\left(\frac{\varepsilon\beta}{\gamma}\right)^t\right),$$

where $\varepsilon = \|\mu\|_2$, $B = \|\theta_0 - \theta_{\mathrm{PS}}\|_2$, and

$$\beta = \max\left\{\max_{x\in\mathcal{X}}\{\|x\|_2^2 + \gamma\}, \max_{(x,y)\in(\mathcal{X},\mathcal{Y}),\theta\in\Theta}\{\sqrt{(x^\top\theta - y + \|x\|_2\|\theta\|_2)^2 + \|x\|_2^2}\}\right\}.$$

Here we take $\mathcal{X} = \{x : \|x\|_2^2 \leqslant 20\}$. Note that the closed form expressions for the update and the performatively stable point in Eq. 7 and Eq. 8 hold for any distribution of $x$ with mean $\mu_x$ and variance $\Sigma_x$, and $\nu$ with mean 0 and variance $\sigma_y^2$. For easier calculation for the smoothness parameter, we truncate the normal distribution of $(x,y)$ such that $\|x\|_2^2 \leqslant 20$. The mean and variance of the resulting truncated distribution can be well approximated by those of the original normal distribution due to the concentration of Gaussian.

We run our experiments on NVIDIA GPUs A100 in a single-GPU setup.

## B.2 Additional Experimental Results

**Ablation study on effects of $\gamma$.** In Figure A1, we compare confidence-region performance under regularization strengths $\gamma = 1$ and $\gamma = 3$. Together with results of $\gamma = 2$ in Figure 1, we can find that as $\gamma$ increases, the gap between the coverage for $\theta_t$ (solid curve) and the bias-adjusted coverage for $\theta_{\mathrm{PS}}$ (dashed curve) vanishes more quickly across iterations. For example, at $t = 3$, the dashed and solid curves are tightly closed for $\gamma = 3$, while a substantial gap remains for $\gamma = 1$. This phenomenon derives that the larger $\gamma$ yields a more strongly convex loss, which both accelerates convergence of the estimate $\widehat{\theta}_t$ to its stable point and reduces the performative bias $\|\theta_t - \theta_{\mathrm{PS}}\|$. Consequently, the bias-awared intervals converge for $\theta_{\mathrm{PS}}$ to the original ones for $\theta_t$ in fewer iterations when $\gamma$ is larger.

**Ablation study on effects of $\varepsilon$.** In Figure A2, we compare confidence-region performance under sensitivity $\varepsilon \approx 0.003$ and $\varepsilon \approx 0.03$. We can find that as $\varepsilon$ increases, the gap between the coverage for $\theta_t$ (solid curve) and the bias-adjusted coverage for $\theta_{\mathrm{PS}}$ (dashed curve) vanishes more slowly across iterations. For example, for $\varepsilon \approx 0.003$, the dashed curves tightly upper-bound the solid curves at $t = 3$, whereas for $\varepsilon \approx 0.03$, a noticeable gap persists even at $t = 5$. This behavior is because a higher $\varepsilon$ amplifies the performative shift (the dependence of the label distribution on $\theta$), which increases the performative bias. That is, stronger sensitivity requires more iterations for $\widehat{\theta}_t$ to approach its stable point, slowing down convergence of the two confidence regions.

**Ablation study on effects of $\sigma_y^2$.** In Figure A4, we compare confidence-region performance under noise level $\sigma_y^2 = 0.1$ and $\sigma_y^2 = 0.4$. We observe that across all settings, PPI with our greedy-selected $\widehat{\lambda}$ is essentially never worse than either baseline $\lambda = 1$ or $\lambda = 0$. When the noise is low ($\sigma_y^2 = 0.1$), greedy $\widehat{\lambda}$ behaves similarly to $\lambda = 0$, placing almost all weight on the true labels. Conversely, when the noise is high ($\sigma_y^2 = 0.4$), greedy $\widehat{\lambda}$ behaves like $\lambda = 1$, relying more heavily on pseudo-labels to reduce variance. For the intermediate noise level $\sigma_y^2 = 0.2$ in Figure 1, greedy $\widehat{\lambda}$ significantly outperforms both baselines by hitting the optimal bias–variance balance.

## B.3 Case Study on Semi-synthetic Dataset

In Section 5, we originally consider experiments on a synthetic dataset because the performative prediction is an on-policy setting, which means we need to collect the corresponding data every time we update the parameter (policy). In all the previous literature on performative prediction, no such dataset is provided. Alternatively, previous work always uses a **semi-synthetic** dataset, which one will need to specify how the data distribution will react and shift according to the new policy.

Following Perdomo et al. [25], we further conduct a case study in a semi-synthetic way on a realistic credit scoring task using a Kaggle dataset [4]. The dataset contains features of individuals and a binary label indicating whether a loan should be granted or not. Consider the setting where a bank uses a logistic regression classifier $\theta$ trained on features of loan applicants to predict their creditworthiness, while the individual applicants respond to this classifier by manipulating their features to induce a positive classification. Following [2], we can formulate this task as performative prediction because applicants' feature distribution $\mathcal{D}(\theta)$ is strategically adapted in response to $\theta$. By applying repeated risk minimization, a performative stable point $\theta_{PS}$ can be achieved.

We treat the data points in the original dataset as the true distribution to compute $\theta_{PS}$. We add Gaussian noise to the original data feature to generate an unlabeled set of the same size. Then, we sample varying $n$ labeled points with $N = 18000$ unlabeled points and perform $t = 5$ repeated risk minimization steps to compute the estimated $\widehat{\theta}_t$ and build the confidence region for it over 100 independent trials. From the experimental results, we find that a coverage of 0.9 is achieved with decreasing width as $n$ increases. Notably, our optimized greedy $\widehat{\lambda}_t$ (orange) achieves the highest coverage and narrowest confidence width compared with when $\lambda$ is fixed to 0 or 1. The results support our proposed theory and strengthen the practical significance of our methods. We hope this case study can inspire future work for practical settings of PPI under performativity.

---

[4]https://www.kaggle.com/c/GiveMeSomeCredit/data

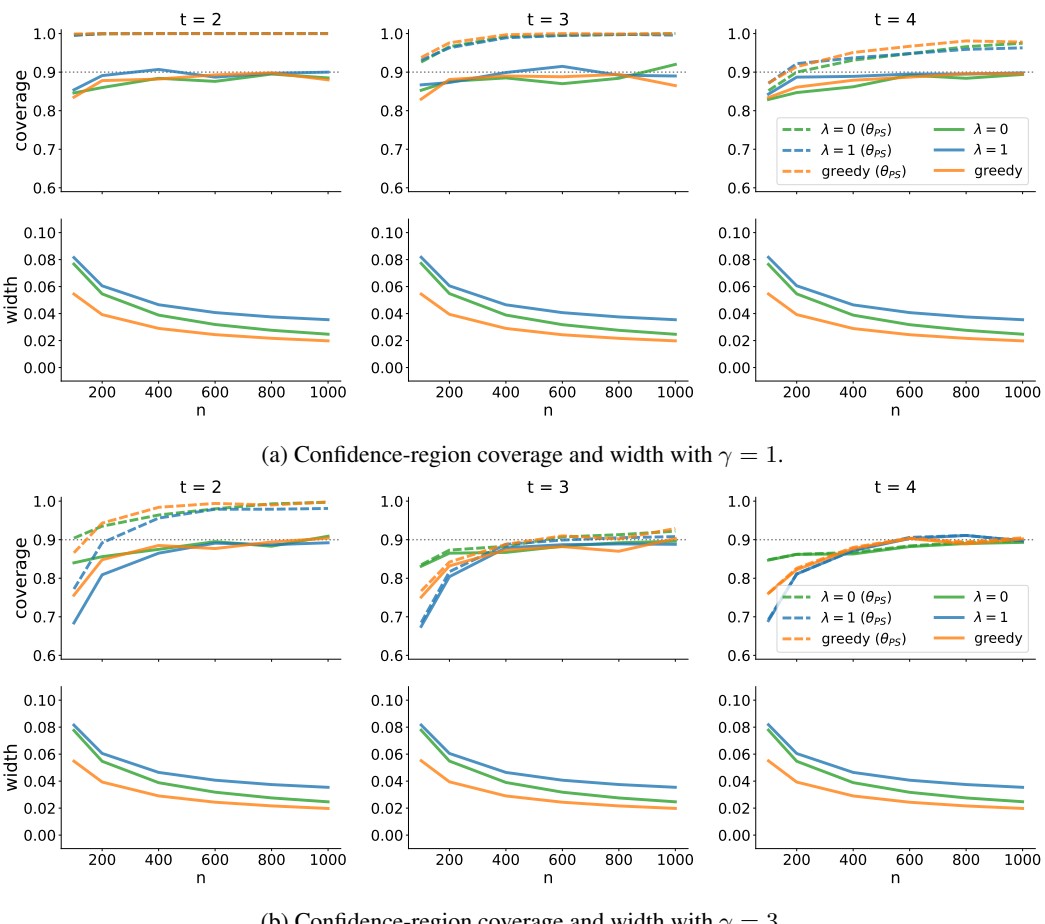

(a) Confidence-region coverage and width with $\gamma = 1$.

(b) Confidence-region coverage and width with $\gamma = 3$.

Figure A1: Confidence-region coverage (top row) and width (bottom row) with different choices of $\lambda$. The setup is the same as in Figure 1, only we change $\gamma = 1$ or $\gamma = 3$.

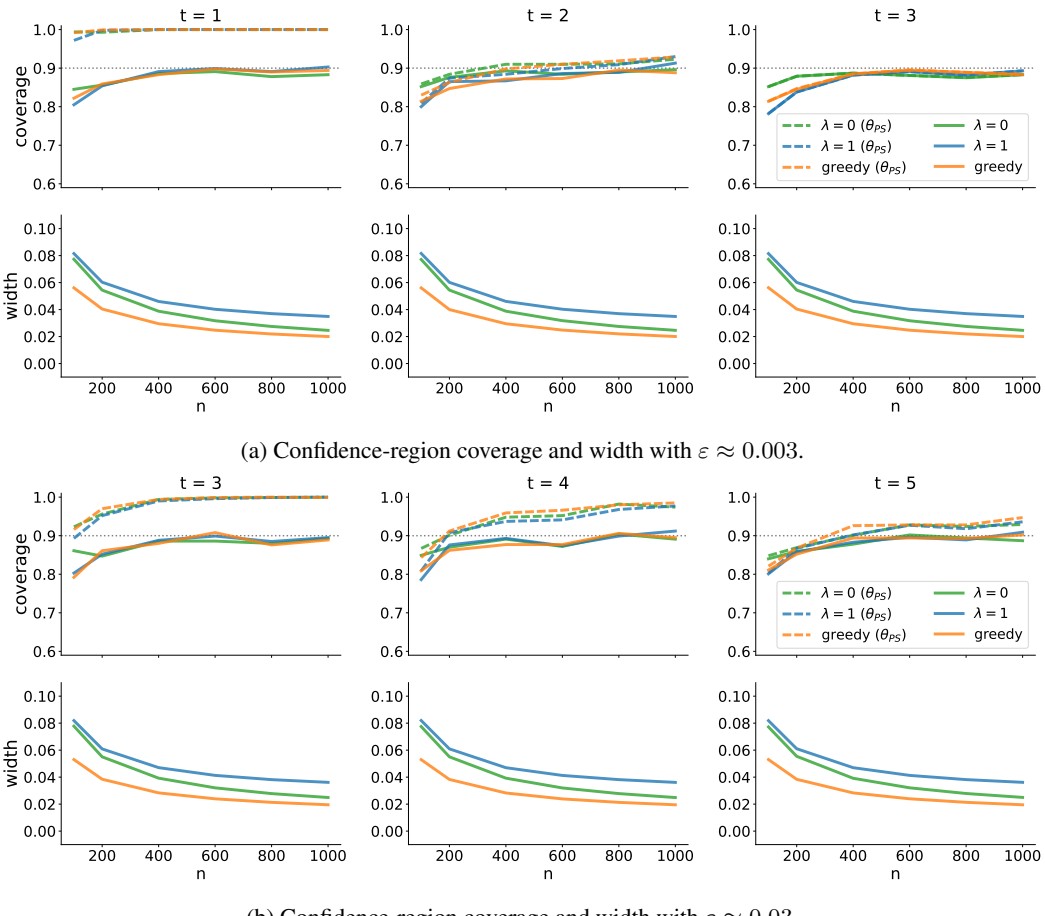

(a) Confidence-region coverage and width with $\varepsilon \approx 0.003$.

(b) Confidence-region coverage and width with $\varepsilon \approx 0.03$.

Figure A2: Confidence-region coverage (top row) and width (bottom row) with different choices of $\lambda$. The setup is the same as in Figure 1, only we change $\varepsilon \approx 0.003$ or $\varepsilon \approx 0.03$.

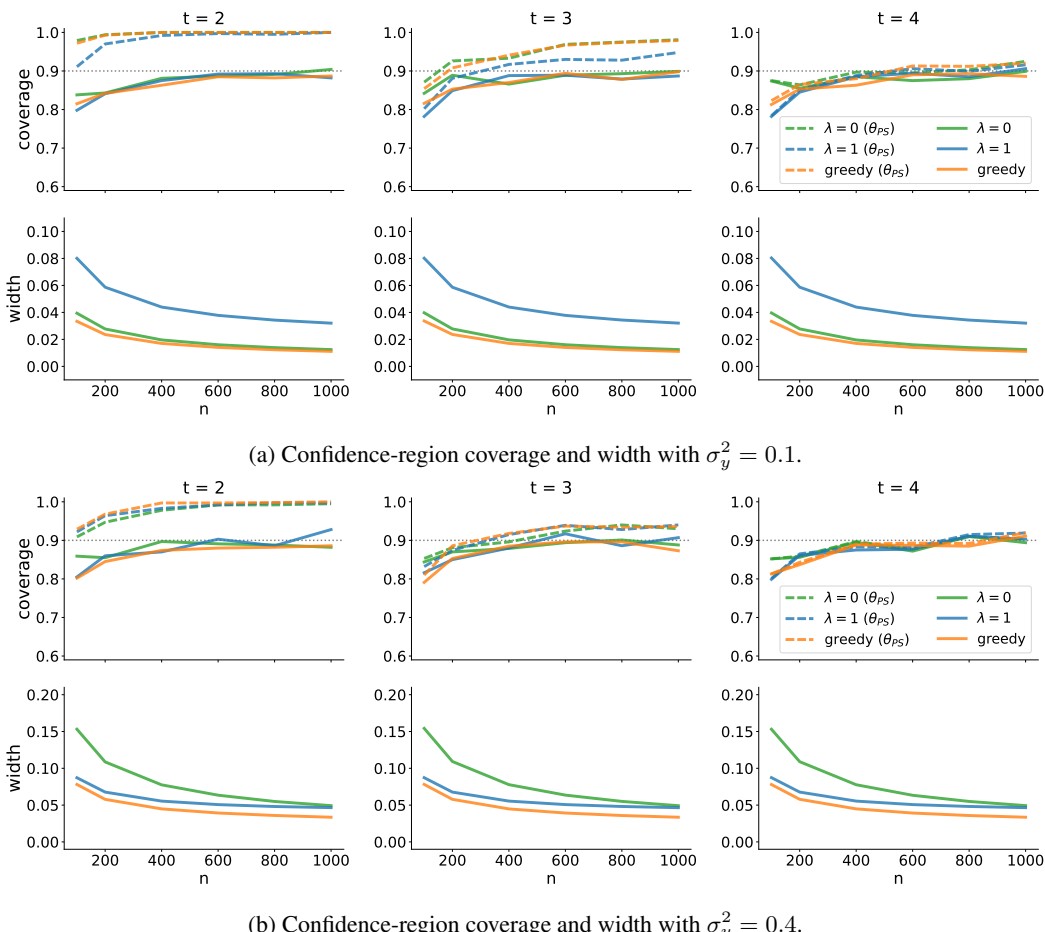

(a) Confidence-region coverage and width with $\sigma_y^2 = 0.1$.

(b) Confidence-region coverage and width with $\sigma_y^2 = 0.4$.

Figure A3: Confidence-region coverage (top row) and width (bottom row) with different choices of $\lambda$. The setup is the same as in Figure 1, only we change $\sigma_y^2 = 0.1$ or $\sigma_y^2 = 0.4$.

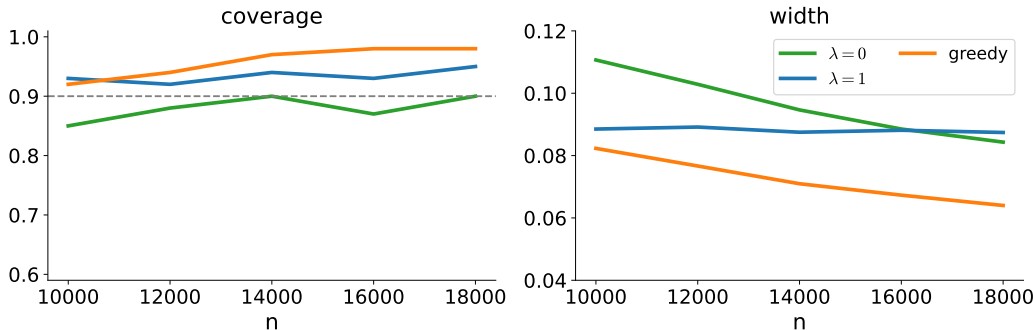

Figure A4: Confidence-region coverage and width for $\theta_t$ ($t = 5$) with different choices of $\lambda$ on the semi-synthetic Kaggle credit scoring dataset.

