# OpenReview forum: "Statistical Inference under Performativity"
_NeurIPS.cc/2025/Conference — NeurIPS 2025 poster_

### Official Review · Reviewer_HBQs · 2025-06-05

**Clarity:** 3
**Significance:** 2
**Originality:** 2
**Rating:** 4
**Confidence:** 3

**Summary:**

The authors establish a central limit theorem for estimates of the performatively stable solution ($\theta_{\text{PS}}$) which are constructed using repeated risk minimization (RRM). Using this central limit theorem, the authors construct confidence regions for $\theta_{\text{PS}}$. Finally, the authors also consider a problem setting where prediction-powered inference (PPI) is used in the presence of performativity. The authors prove a similar central limit theorem and a corresponding estimate for the optimal weights between ``gold-standard" labeled data and the annotated data.

**Questions:**

* Can you provide non-synthetic experiments? How do the constructed confidence intervals perform on non-synthetic data?

**Ethical Concerns:**

["NO or VERY MINOR ethics concerns only"]

**Final Justification:**

The authors agreed to include relevant work, and addressed my concerns over all synthetic experiments.

**Limitations:**

* Consider adding a limitations section to the main text

**Quality:**

2

**Strengths And Weaknesses:**

**Strengths**
* The results on selecting optimal $\{\lambda_t\}$ for performative PPI are interesting and potentially useful.
* The paper develops asymptotic results for a generic $t \in [T]$ as $n$ grows; this is distinct from prior work on asymptotics of performative estimates.

**Weaknesses**

* The authors are missing a comparison with the closely related work [1]; this work provides asymptotic results for the repeated stochastic gradient descent estimator of $\theta_{\text{PS}}$. The asymptotic results here are in the regime $n=1, T \rightarrow \infty$ so the results in the submission are distinct, but discussion of this work is still needed. For example, how are the proof techniques in this work novel to those in [1]? Also, what are the motivations for this asymptotic setting? It seems somewhat less realistic then what is considered in [1].

* The claim that 'the work initiates the study of inference in performativity' is not true; see, for example, [1], [2]. This is repeated throughout the paper and re-writes are needed to eliminate this.

* The theory does not account for any optimization error in the risk minimization step of RRM

* Use of the constructed confidence intervals requires a density estimation step where model $M(z, \theta; \psi)$ is used to estimate the performative distribution $p(z, \theta)$. Furthermore, the theory requires an assumption of perfect convergence on the optimization over $\psi$.

* All experiments are on simulated data; I think this is an issue for the PPI experiments in particular. I am not aware of practical settings which both require the use of PPI and have performative aspects. The authors adding a data set where this is plausibly the case would greatly strengthen the results.

**Minor comments**

* 'Lastly,' repeated twice in the notation section
* Line 53 'perforamtive'


[1] Cutler et al. Stochastic Approximation with Decision-Dependent Distributions: Asymptotic Normality and Optimality. https://arxiv.org/abs/2207.04173

[2] Bracale et al. Micro-foundation inference in strategic prediction. https://arxiv.org/abs/2411.08998

---

> ### Author Rebuttal · Authors · 2025-07-30
>
> Thank you for your valuable comments. We will revise our paper according to your helpful suggestions. But we think there exists some misunderstanding. Below, we address your questions point by point and hope that our answers can ease your concerns.
>
> **Q1:** The authors are missing a comparison with the closely related work [1]. The claim that "the work initiates the study of inference in performativity" is not true.
>
> **A1:** Thank you for providing the related papers for us. We will discuss and cite them in our future revision.
>
> However, we acknowledge that [1] establishes the asymptotic normality and minimax optimality for performative settings. While both [1] and our work study the asymptotic normality of performative estimators, the problem setups and theoretical focus differ a lot. More importantly, **[1] does not provide an end-to-end inference framework** (please see further discussions below). Therefore, we respectfully disagree that the statement of "the work initiates the study of inference in performativity" is not true. But we can totally understand that our claim can cause confusion at the first glance, and we will modify in our future revision to clarify that we initiate the inference framework for repeated risk minimization in the batch setting under performativity.
>
> 1) Setup: [1] focuses on stochastic gradient update with one sample, whereas our work analyzes the empirical risk minimizer at each round.
>
> 2) Theory: [1] establishes only the normality result. Their asymptotic covariance involves the gradient of the density function $p$ in their $\nabla R(x^*)$, as in our $\nabla G(\theta_k)$. However, they only provide examples assuming a parametric form of $p$ and treat all the structural knowledge as given. Thus, they don't propose any covariance estimation method from data. In contrast, we further deal with the density function and provide an end-to-end inference method for constructing confidence intervals. All the theoretical results are validated by extensive experiments.
>
> 3) Technical details: Our analysis draws on M-estimation techniques from empirical process theory, while [1] follows the literature on stochastic optimization.
>
> As the reviewer noted, their setting is analogous to a stochastic approximation of our model with $n=1, T\to \infty$.
> Our model adopts a “batch” formulation that **better reflects practical scenarios**. For instance, in policy-making, policies are often updated based on population-level feedback, which is often collected through surveys. These surveys often suffer from non-response issues, so we extend the PPI framework to help address the data scarcity issue.
>
> For [2], it studies identifiability and estimation error under a specific microfoundation model with performativity. They did not discuss about confidence interval construction. In contrast, our main contribution is establishing an end-to-end inference method for constructing confidence intervals, which is missing in existing literature, to the best of our knowledge. We will clarify that in the related work section.
>
> To sum up, we will revise the phrasing to more accurately reflect the existing literature, and will add citations [1][2] and comparisons with the works you mentioned.
>
> **Q2:**
> The theory does not account for any optimization error in the risk minimization step of RRM.
>
> **A2:**
> While our theoretical analysis focuses on the empirical risk minimizer, the convexity of the loss function allows for incorporating optimization error with standard guarantees.
> For instance, if the loss function $\ell(z,\theta)$ is $\beta$-smooth and $\gamma$ strongly convex w.r.t. $\theta$, a gradient descent with $M$ iterations with an initialization $\theta_{\text{init}}$ and a learning rate $0<\eta\leq \frac{1}{\beta}$ guarantees that the distance between the gradient descent iterate $\hat{\theta}_1^{\text{opt}}$ and and the empirical risk minimizer $\hat{\theta}_1$ as follows (we refer to [3] for more details)
>
> $$
> \\|\hat{\theta}_1^{\text{opt}}-\hat{\theta}_1\\|_2^2\leq \frac{8\eta\beta^2}{\gamma^2}(1-\frac{\gamma}{\beta})^{M-1}\mathcal{L},
> $$
>
> where $\mathcal{L}=\frac{1}{n}\sum_{i=1}^{n}\ell (z_{0,i};\theta_{\text{init}}).$
>
>
> In practice, gradient-based methods with a sufficient number of iterations can be used to ensure the optimization error remains desirable. We will discuss this part more in our revision.
>
> **Q3:**
> Use of the constructed confidence intervals requires a density estimation step where model $M$ is used to estimate the performative distribution $p(z,\theta)$. Furthermore, the theory requires an assumption of perfect convergence in the optimization over $\psi$.
>
> **A3:** As our goal is to provide an end-to-end inference method, it's a necessary step to obtain an accurate estimation of the density (gradient) function and thus the variance term. The accuracy of this step depends on the structure of the distribution map and the estimation method used.
>
> Regarding the assumption of perfect convergence over $\psi$, it's a simplifying assumption that the model class $M$ is rich enough to approximate the true density. In practice, our optimization procedure achieves high accuracy, as demonstrated in our experimental results.
>
>
> **Q4:**
> All experiments are on simulated data ... The authors adding a data set where this is plausibly the case would greatly strengthen the results.
>
> **A4:**
> Thanks for the suggestion. We originally only considered experiments on a synthetic dataset because the performative prediction is an on-policy setting, which means we need to collect the corresponding data every time we update the parameter (policy). In all the previous literature on performative prediction, no such dataset is provided. People always use a **semi-synthetic** dataset, which people will need to specify how the data distribution will react and shift according to the new policy.
>
> In order to ease your concern, we further conduct a case study in a semi-synthetic way following previous literature on a realistic credit scoring task using *Give me a credit* dataset (due to NeurIPS policy, we cannot include links, but this is a public dataset online). The dataset contains features of individuals and a binary label indicating whether a loan should be granted or not. Consider the setting that a bank uses a logistic regression classifier $\theta$ trained on features of loan applicants to predict their creditworthiness, while the individual applicants respond to this classifier by manipulating their features to induce a positive classification. Following [4], we can formulate this task as performative prediction because applicants' feature distribution $\mathcal{D}(\theta)$ is strategically adapted in response to $\theta$. By applying repeated risk minimization, a performative stable point $\theta_{PS}$ can be achieved.
>
> We treat the data points in the original dataset as the true distribution to compute $\theta_{PS}$. We sample varying $n$ labeled points with $N$ unlabeled points and perform $t=5$ repeated risk minimization steps to compute the estimated $\hat{\theta}_t$ and build the confidence region for it over 100 independent trials. From the experimental results, we find that a coverage of 0.9 is achieved as the width decreases with increasing $n$. The results support our proposed theory and strengthen the practical significance of our methods. We will complete this case study and include it in our final version. We hope this case study can inspire future work for practical settings of PPI under performativity.
>
> | $n$      | 10k   | 12k   | 14k   | 16k   | 18k   |
> | :------- | :---- | :---- | :---- | :---- | :---- |
> | coverage | 0.85  | 0.88  | 0.90  | 0.87  | 0.90  |
> | width    | 0.110 | 0.102 | 0.094 | 0.088 | 0.084 |
>
>
>
>
>
> **Q5:**
> Minor comments.
>
> **A5:**
> Thank you for your careful examination. We will change the fix them in our revision.
>
>
> [1] Cutler et al. "Stochastic Approximation with Decision-Dependent Distributions: Asymptotic Normality and Optimality."
>
> [2] Bracale et al. "Micro-foundation inference in strategic prediction."
>
> [3] Polyak. "Gradient methods for the minimisation of functional."
>
> [4] Perdomo et al. "Performative prediction."

---

> > ### Comment · Reviewer_HBQs · 2025-08-01
> >
> > Thank you to the authors for thoroughly addressing my points. I agree with them that [1], [2] are quite different from their work. I thank them for agreeing to include citations of these, but I still argue that claims of  "initiating the study of inference in performativity" should be relaxed. Given the additional experiments and clarifications I will raise my score to a 4.

---

> > > ### Author Response · Authors · 2025-08-01
> > > **Thanks for your kind response and willingness to raise the score**
> > >
> > > Dear reviewer HBQs,
> > >
> > > We appreciate your further feedback and willingness to raise the score. We will relax the statement of "initiating the study of inference under performativity" to "initiating the inference framework for repeated risk minimization in the batch setting under performativity". We will further discuss the relationship to work [1] and [2] in detail in our revision.

---

### Official Review · Reviewer_BWhj · 2025-07-02

**Clarity:** 3
**Significance:** 3
**Originality:** 3
**Rating:** 5
**Confidence:** 5

**Summary:**

This paper studies how to perform statistical inference when a predictive model influences the data-generating process — a phenomenon known as *performativity*.
At each round the learner deploys a model parameter \\( \\theta_t \\), new data \\( z_{t,i} \\sim D(\\theta_t) \\) are generated in response, and the learner updates the model by empirical risk minimisation.
Under standard smoothness, convexity, and a Lipschitz condition linking \\( D(\\theta) \\) to \\( \\theta \\), the authors show that the sequence \\( \\theta_t \\) converges linearly to a unique performatively stable point \\( \\theta_{\\mathrm{PS}} \\).

The main theoretical contribution is a central limit theorem: for fixed \\( t \\) and sample size \\( n \\), one has
\\[
\\sqrt{n}\\,(\\hat\\theta_t - \\theta_t) \\; \\xrightarrow{d} \\; \\mathcal{N}(0, V_t),
\\]
where the asymptotic covariance \\( V_t \\) depends on a product of Jacobians \\( \\nabla G(\\theta_k) \\) and a sandwich matrix built from the score of the unknown density \\( p(z, \\theta) \\).
Because \\( p \\) is not observed, the authors propose estimating its score by fitting a flexible surrogate density \\( M(z, \\theta; \\psi) \\) with a score-matching objective augmented by small “policy perturbations” in \\( \\theta \\).
Combining the CLT with the geometric convergence of \\( \\theta_t \\) yields asymptotically exact confidence regions for the stable point \\( \\theta_{\\mathrm{PS}} \\).
The framework is further extended with *prediction-powered inference*: abundant pseudo-labels are blended with scarce ground-truth labels to reduce variance, and a second CLT quantifies the resulting efficiency gains.
Simulations in a performative linear-regression setting confirm nominal coverage of the confidence sets and illustrate how the pseudo-label weight improves precision.

**Questions:**

- **Meaning of \\( p(z, \\theta_k) \\).**
  I understand that \\( p(z, \\theta_k) \\) refers to the probability density (or mass) function of the distribution \\( D(\\theta_k) \\) from which the data at round \\( k \\) are drawn.
  A brief reminder of this definition in the main text might help readers.

- **Missing of appendix.**
  I understand that not all assumptions and proofs can be included in the main text.
  However, the appendix containing these details is missing from the submission, which makes the paper difficult to understand on its own.

- **Precision of the variance estimates.**
  Estimating \\( \\nabla G(\\theta_k) \\) and the covariance \\( V_t \\) requires approximating an intricate score function by the surrogate \\( M(z, \\theta, \\psi) \\).
  Because this involves sums and products over all past rounds, a discussion of finite-sample error — perhaps via non-asymptotic bounds or sensitivity analysis — would strengthen the practical relevance of the method.

- **Dependence on the model class \\( \\mathcal{D}(\\tilde{\\theta}) \\).**
  The accuracy of the gradient estimates naturally depends on how well \\( M \\) can approximate the true family \\( \\{ \\mathcal{D}(\\theta) \\} \\).
  I would appreciate comments on how the complexity of this family affects approximation error, and whether model-selection guidance could be offered.

- **Robustness to annotation error in experiments.**
  The numerical study assumes an annotator that is almost unbiased, with only a small mean shift.
  In many real applications, annotation models may be substantially misspecified.
  Exploring scenarios with larger mean or variance shifts, or other structural errors, would give readers a clearer sense of the method’s robustness.

**Ethical Concerns:**

["NO or VERY MINOR ethics concerns only"]

**Final Justification:**

Dear AC and authors,

I would like to recommend an acceptance to the paper.

Best wishes,
Your reviewer

**Limitations:**

This is the same the last three questions in the questions

- **Precision of the variance estimates.**
  Estimating \\( \\nabla G(\\theta_k) \\) and the covariance \\( V_t \\) requires approximating an intricate score function by the surrogate \\( M(z, \\theta, \\psi) \\).
  Because this involves sums and products over all past rounds, a discussion of finite-sample error — perhaps via non-asymptotic bounds or sensitivity analysis — would strengthen the practical relevance of the method.

- **Dependence on the model class \\( \\mathcal{D}(\\tilde{\\theta}) \\).**
  The accuracy of the gradient estimates naturally depends on how well \\( M \\) can approximate the true family \\( \\{ \\mathcal{D}(\\theta) \\} \\).
  I would appreciate comments on how the complexity of this family affects approximation error, and whether model-selection guidance could be offered.

- **Robustness to annotation error in experiments.**
  The numerical study assumes an annotator that is almost unbiased, with only a small mean shift.
  In many real applications, annotation models may be substantially misspecified.
  Exploring scenarios with larger mean or variance shifts, or other structural errors, would give readers a clearer sense of the method’s robustness.

**Quality:**

3

**Strengths And Weaknesses:**

Very good results and answering important questions. Clearly presented.
I commend the work and consider it a valuable contribution. However, I am giving a borderline accept due to concerns regarding the **precision of the variance estimates** (detailed below).
In my view, this issue is central: if the estimates prove unreliable or highly sensitive in finite samples, the practical value of the results would be significantly diminished.
On the other hand, if the variance estimation procedure is shown to perform well, either through theoretical guarantees or empirical validation, I would view the work as a very significant contribution to the literature and would be inclined to change my rating to **strong accept**.

---

> ### Author Rebuttal · Authors · 2025-07-30
>
> Thank you for your valuable comments and for finding our paper meaningful. Below, we address your concerns point by point. Hope that our answers can further ease your concerns.
>
> **Q1:**
> A brief reminder of the definition of $p(z,\theta)$ in the main text might help readers.
>
> **A1:** Thank you for your suggestions on the presentation. We will include reminders and more explanations for readers in our revision.
>
>
> **Q2:**
> Missing of appendix.
>
> **A2:**
> We are so sorry to cause confusion. But **we indeed include our appendix in the submission**. We did not attach it directly following the paper, but our appendix is included in the supplementary material. Both ways are lawful and stated in the guidance of the NeurIPS submission.
>
> **Q3:**
> Variance estimates: finite-sample error, perhaps via non-asymptotic bounds or sensitivity analysis would strengthen the practical relevance of the method ... I would appreciate comments on how the complexity of this family affects approximation error, and whether model-selection guidance could be offered.
>
> **A3:**
> Thank you for raising this point. We agree with your assessment of the estimation error. While important, addressing this issue is beyond the primary scope of our current work because **this will require case-by-case assumptions** (see below for an example).
> Our main result is the consistency guarantee of the estimator and an end-to-end method for confidence interval construction. Depending on the specific distribution map family (e.g., location-scale family) and the estimation method used (e.g., regression, score matching), different estimation error guarantees could apply.
>
> **A concrete example.**  Consider a specific parametric form of $\mathcal{D}(\theta)$: the location-scale family defined as $z_\theta\sim \mathcal{D}(\theta) \iff z_\theta = z_0+\mu \theta$, where $z_0$ is from a base distribution, $\mu\in \mathbb R^{m\times d}$ is an unknown parameter, and $m,d$ are the dimensionalities of $z,\theta$, respectively. In the specific parametric model for $\mathcal{D}(\theta)$, the parameter $\mu$ can be estimated by least square regression using all samples collected from $T$ iterations, where at the $i$-th iteration, samples are drawn for parameter $\hat{\theta}_i$.
>
> Let $\hat{\Theta}=(\hat{\theta}_1,\cdots,\hat{\theta}_1,\cdots, \hat{\theta}_T, \cdots,\hat{\theta}_T)^\top  \in \mathbb{R}^{nT\times d}$ denote the augmented sample-wise design matrix, and $\Theta=(\theta_1,\cdots,\theta_T)^\top \in \mathbb{R}^{T\times d}$ denote the population-wise design matrix. Then standard concentration bounds imply that
>
> $\\|\mu - \hat{\mu}\\|_{\text{op}} $
>
> $=\\| (\hat{\Theta}^\top \hat{\Theta})^{-1} (\hat{\Theta}^\top Z_0) \\|_{\text{op}}$
>
> $\leq \frac{1}{\lambda_{\min}(\hat{\Theta}^\top \hat{\Theta})} \\|\hat{\Theta}^\top Z_0\\|_{\text{op}}$
>
> $\lesssim \frac{1}{n \lambda_{\min}(\Theta^\top \Theta)} \sqrt{n T (d + m)},$
>
>
> with high probability, given that $n$ is sufficiently large and $\{\theta_i\}$ span the whole space. In the second inequality, the operator norm bound follows from the standard $\epsilon$-net arguments and Bernstein-type inequality, and the eigenvalue perturbation is controlled using Weyl's inequality.
>
>
> **Q4:**
> Robustness to annotation error in experiments: Exploring scenarios with larger mean or variance shifts, or other structural errors, would give readers a clearer sense of the method’s robustness.
>
> **A4:** We refer the reviewer to Appendix B.2 in the supplementary material, where we conduct ablation studies on the effects of different choices of regularization strengths $\gamma$, sensitivity (mean) $\varepsilon$, and noise level (variance shift) $\sigma_y^2$. The results shown in Figures A1-A3 suggest the robustness of our proposed methods.

---

> > ### Comment · Reviewer_BWhj · 2025-08-08
> > **Thank you for the rebuttal**
> >
> > I appreciate the effort that the authors spent on providing new theoretical results on finite sample guarantees. I find it is interesting. Together with the theoretical interest of this work,I would recommend to accept the paper and raise my score to 5

---

> > > ### Author Response · Authors · 2025-08-09
> > > **Thanks for your kind response and willingness to raise the score**
> > >
> > > Dear reviewer BWhj,
> > >
> > > We appreciate your further feedback and willingness to raise the score to 5! We really appreciate your insightful comments and suggestions. We will incorporate the finite sample results in our future revision.

---

> ### Author Response · Authors · 2025-08-03
> **Further discussion on variance estimation**
>
> We are grateful to the reviewer for raising the point regarding the importance of precise variance estimation. To address this, we would like to elaborate further and provide **new theoretical results** on variance estimation and highlight our experimental evidence a bit more.
>
>
> **Theory:** Originally, we provided a consistency guarantee for the variance estimator. However, finite-sample guarantees can depend on specific assumptions. In many cases, strong estimation guarantees can be established under mild assumptions, as demonstrated in the concrete example provided above. We further notice that we actually can address this problem further than the argument we provided in the appendix because we can relax to requiring $L_2$-convergence in the process score matching instead of asking for optimizer convergence. We have now extended this to **a more general theoretical result**, described as follows at a high level because of limited space. We will include it in the future revision.
>
> Recall that $J(\psi,\theta)$ and $\hat J_{n,k}(\psi,\theta)$ denote the score matching objective and its empirical counterpart. Assume that $s(z,\theta;\psi)=\nabla_\theta \log M(z,\theta;\psi)$ is rich enough to express $\nabla_\theta \log p(z,\theta)$, and that the optimization error vanishes in the sense that $\hat J_{n,k}(\hat\psi(\theta),\theta)\leq \min_{\psi\in\Psi}\hat J_{n,k}(\psi,\theta)+a_n$ where $a_n=o(1)$. Under additional boundedness and uniform entropy conditions, we establish the following bound using Dudley's inequality and Taylor expansion:
> $$\mathbb E_\theta \left[\\| \nabla_\theta \log p(z;\theta)-s(z,\theta,\hat{\psi}(\theta))\\|^2 | \hat{\psi}(\theta) \right]= O_p\left( \frac{1}{\sqrt{n}}+\frac{1}{\sqrt{\min(n,k)}}\frac{1}{\eta} + \eta + a_n \right).$$
> By decomposing the error between the empirical and true gradient estimates into several components, where each varies in whether it uses empirical or population quantities, we then obtain that
> $$\\|\hat{G}_k-\nabla G(\theta_k)\\|= O_p\left( \frac{1}{\sqrt{n}}+\frac{1}{\sqrt{\min(n,k)}}\frac{1}{\eta} + \eta + a_n \right).$$
> This ensures the precision of the variance estimates under a suitable choice of $\eta$ and optimization method.
>
> **Experiment:** In practice, for model selection of $M$, we recommend using neural networks as a foundation due to their strong expressiveness and ease of optimization. In Figure 3 of the main article, we evaluate the estimation quality of two score-matching estimators. We find that the DNN estimator can achieve $J(\psi,\theta)<0.01$ over varying $n$ and $t$, indicating the perfect approximation of our learned model $M(z,\theta;\psi)$ to the true $p(z,\theta)$. Also, the variance-estimation error remains negligible and decreases as $n$ grows, verifying the feasibility of using our score matching models to fit $\nabla \theta \log p(z, \theta)$ for estimating $\nabla G(\theta_k)$ and $V_t$. We provided multiple synthetic experiments in the main article and supplementary material using the DNN-based estimator, varying the mean and noise levels for robustness checks and sensitivity analysis.
> We further conduct a case study in a semi-synthetic way, following previous literature on a realistic credit scoring task. Specifically, we conduct a case study on a realistic credit scoring task using *Give me a credit* dataset. From the experimental results, we find that a coverage of 0.9 is achieved with decreasing width as $n$ increases. The results support our proposed theory and strengthen the practical significance of our methods.
>
>
> We appreciate your recognition of our paper's significance. We hope these explanations help address your concerns.

---

> > ### Author Response · Authors · 2025-08-07
> >
> > Dear Reviewer BWhj,
> >
> > Thank you again for your time and valuable comments on our paper. We truly appreciate your constructive comments and contributions to the review process.
> >
> > As the NeurIPS discussion period ends tomorrow (Aug 8, 11.59pm AoE), we kindly ask whether you have further concerns and questions.
> >
> > We have further addressed the variance estimation question by adding additional theoretical results, and also provided more on the evidence of our experimental results. Please let us know if you have any concerns or require further clarification.
> >
> > Respectfully,
> >
> > Authors of paper submission 17654

---

### Official Review · Reviewer_CPwq · 2025-07-03

**Clarity:** 3
**Significance:** 2
**Originality:** 2
**Rating:** 4
**Confidence:** 3

**Summary:**

This paper initiates the study of statistical inference under performativity, where prediction-informed decisions influence the outcome variable, leading to a feedback loop between predictive models and the data-generating process. The authors focus on the performative stable point and propose a framework for valid inference in this dynamic setting. The main contributions are (1) the derivation of a central limit theorem for estimators obtained through repeated risk minimization (RRM), enabling confidence region construction for the performative stable parameter, and (2) an extension of prediction-powered inference (PPI) methods to performative settings, leveraging unlabeled data and machine learning predictions to obtain more efficient inference. Theoretical results are validated through synthetic experiments.

**Questions:**

1. Would the framework extend to non-convex loss functions or to settings where D(θ) exhibits more complex dependence on θ?
2. Are there diagnostics that practitioners can use to check whether the model is correctly capturing performativity effects in practice?

**Ethical Concerns:**

["NO or VERY MINOR ethics concerns only"]

**Final Justification:**

I recommend a borderline accept, as I find this to be a theoretically solid paper. However, I did not assign a higher score because I still have concerns about the method’s applicability. In the authors’ response, they indicated that no real data can be used and that all applicable datasets are semi-synthetic. As a result, I remain unsure about the intended target audience for the method.

**Limitations:**

See weakness.

**Quality:**

3

**Strengths And Weaknesses:**

Strengths:

The paper tackles the statistical inference in performative prediction, which is relevant for domains where model-informed decisions recursively shape the data (e.g., economics, policy, and social sciences). The theory is solid.

Weaknesses:

1. Motivation: The motivation of this paper is unclear to me. A more realistic example where the methods could work will be helpful. Otherwise, it seems to be a simple combination of PPI and Performative prediction.

2. Empirical Scope: The experimental validation is currently limited to synthetic linear models. Application to more complex or realistic settings (e.g., non-linear models or real-world feedback systems) would further strengthen the practical significance of the proposed methods. Real data application will also be important.

3. Practical Implementation: The paper introduces novel estimation strategies (e.g., policy perturbation for score matching), but more details or discussion on computational aspects, tuning, or diagnostics for these methods would make the framework more accessible for practitioners.

---

> ### Author Rebuttal · Authors · 2025-07-30
>
> Thank you for your comments. Below, we address your concerns point by point. We hope that our answers can further ease your concerns and raise our scores.
>
> **Q1:**
> Motivation: A more realistic example where the methods could work will be helpful.
>
> **A1:** To clarify, there are two significant contributions motivated by real applications in this paper.
>
> First, as mentioned in our paper in Section 3, we initiate the study of the inference perspective of the celebrated repeated risk minimization (RRM) algorithm for performative prediction. This algorithm has already been widely used in applications regarding performativity, but previous work did not provide inference results like asymptotic normality and confidence region, etc., which makes the obtained parameter (policy) lack uncertainty quantification. When it comes to policy, only knowing that the algorithm will converge to a stable policy is not enough; we need to provide the concrete parameters as well as error bars incurred by randomness and optimization. So, that motivates the whole Section 3. To give a concrete, realistic example of the setting in Section 3, one can think of applications relevant to domains where model-informed decisions recursively shape the data (e.g., economics, policy, and social sciences). Indeed, in policy-making contexts, such as setting tax rates or credit score thresholds, policymakers often adjust policies dynamically in response to observed outcomes, which in turn influence the behavior of many agents.
>
> Second, people often conduct surveys to collect information and feedback for policy development. As noted by [1], these surveys often suffer from non-response issues. This highlights the importance of addressing data scarcity in the performative setting, which motivates our Section 4 that considers how to derive a principled method to use unlabeled data to improve the estimation and inference under performativity. The principled method (PPI) is **based on the asymptotic normality derived in Section 3, which is a novel result itself**. We want to emphasize that the normality results are highly non-trivial to obtain. In Section 4, we extend the whole PPI framework in the performative setting, which is our other contribution.
>
>
> **Q2:**
> Empirical Scope: Application to more complex or realistic settings would further strengthen the practical significance of the proposed methods.
>
> **A2:** Thanks for the suggestion. We originally only considered experiments on a synthetic dataset because the performative prediction is an on-policy setting, which means we need to collect the corresponding data every time we update the parameter (policy). In all the previous literature on performative prediction, no such dataset is provided. People always use a **semi-synthetic** dataset, which people will need to specify how the data distribution will react and shift according to the new policy.
>
> In order to ease your concern, we further conduct a case study in a semi-synthetic way following previous literature on a realistic credit scoring task using *Give me a credit* dataset (due to NeurIPS policy, we cannot include links, but this is a public dataset online). The dataset contains features of individuals and a binary label indicating whether a loan should be granted or not. Consider the setting that a bank uses a logistic regression classifier $\theta$ trained on features of loan applicants to predict their creditworthiness, while the individual applicants respond to this classifier by manipulating their features to induce a positive classification. Following [2], we can formulate this task as performative prediction because applicants' feature distribution $\mathcal{D}(\theta)$ is strategically adapted in response to $\theta$. By applying repeated risk minimization, a performative stable point $\theta_{PS}$ can be achieved.
>
> We treat the data points in the original dataset as the true distribution to compute $\theta_{PS}$. We sample varying $n$ labeled points with $N$ unlabeled points and perform $t=5$ repeated risk minimization steps to compute the estimated $\hat{\theta}_t$ and build the confidence region for it over 100 independent trials. From the experimental results, we find that a coverage of 0.9 is achieved as the width decreases with increasing $n$. The results support our proposed theory and strengthen the practical significance of our methods. We will complete this case study and include it in our final version. We hope this case study can inspire future work for practical settings of PPI under performativity.
>
> | $n$   | 10k   | 12k   | 14k   | 16k   | 18k   |
> |:------|:------|:------|:------|:------|:------|
> | coverage | 0.85  | 0.88  | 0.90  | 0.87  | 0.90  |
> | width    | 0.110 | 0.102 | 0.094 | 0.088 | 0.084 |
>
>
> **Q3:**
> Practical Implementation: More details or discussion on computational aspects, tuning, or diagnostics for these methods would make the framework more accessible for practitioners. Are there diagnostics that practitioners can use to check whether the model is correctly capturing performativity effects in practice?
>
> **A3:** We indeed include more details in the appendix. We refer the reviewer to Appendix B.1 in the supplementary material for additional experimental details on computational aspects and tuning of the proposed method.
>
>
> **Q4:**
> Would the framework extend to non-convex loss functions or to settings where $\mathcal{D}(\theta)$ exhibits more complex dependence on $\theta$?
>
> **A4:**
> As we point out in Remark 3.2, strong convexity of the loss function $\ell(z,\theta)$ and sensitivity of the distribution map $\mathcal{D}(\theta)$ are necessary to guarantee the convergence of the trajectory $\\{\theta_t\\}_{t\geq 0}$.
> 1) If the loss function is convex but not strongly convex, RRM may fail to converge. Additionally, if the loss function is non-convex, the optimizer may not even be unique, which makes doing statistical inference pointless.
> 2) The conditions of $\mathcal{D}(\theta)$ are mild enough to contain a wide range of cases where the distribution admits a density function $p(z,\theta)$.
>
>
>
> [1] Huang et al. "Insufficient effort responding: examining an insidious confound in survey data."
>
> [2] Perdomo et al. "Performative prediction."

---

> > ### Comment · Reviewer_CPwq · 2025-08-06
> >
> > I thank the reviewers for their detailed responses in the rebuttal. However, I still find it difficult to understand the practical utility of the proposed method. For example, in the original PPI paper, the authors demonstrated potential applications through several real-world examples, such as AlphaFold, gene expression prediction, and galaxy classification, which made the method’s relevance more convincing. In contrast, this paper does not provide a concrete, real-world example of how the method could be applied. All experiments are based on simulations, which limits the demonstration of its practical impact.
> >
> > I would appreciate it if the authors can provide several concrete examples on how their method can be applied. For example, what are the real-world tasks, what is the labeled data, and what is the unlabeled data.

---

> ### Author Response · Authors · 2025-08-07
> **Further Answers to Reviewer CPwq**
>
> We thank the reviewer for getting back to us and ask further questions. We will try to clarify with further details about a more concrete example: the credit score example with **real** dataset that we **conduct additional experiments on** (please see our initial rebuttal for our additional experimental results.)
>
> Consider a bank has a policy to predict which person is qualified to get a loan. Applying this policy to the population, this could change the features like consumption habit of the population in the long term --- people know that if they spend all their salary in the first couple of days will make them hard to get a loan, so they will be spending money in a more conservative way, or sometimes, even manipulate their consumption pattern in order to get higher chance to get a loan. In this example, we can see that **the policy is affecting the feature distribution of the population**, which means, **the target distribution to predict is affected by the prediction itself**. Here, the policy can be viewed as the parameter of a prediction model, so in contrast to regular supervised learning, the aim of performative prediction is to optimize: $\min_\theta E_{(x,y)\sim\mathcal{D}(\theta)}\ell(x,y,\theta).$ In this setting, x is the feature, and y is the label about the undying truth --- whether the person is qualified and pay back the loan or not. In practice，we usually can only obtain a random subset of labelled data for privacy concerns or high labeling budget. Besides, one interesting special case is strategic learning --- a person could manipulate his/her features to fool the predictor if they already know the policy, for instance, one can fake a bit on the salary level and also change their consumption habit, but the underlying y is not changed. This is well-known in Econ CS as strategic manipulation.
>
> In our paper, **as our first contribution**,  which is also the main contribution of our paper, we provide inference for the above optimization problem. That means that we can provide uncertainty quantification for the stable point of the above optimization. This can allow us to build error bars and conduct hypothesis testing etc. and can know "how far" is the obtained policy by using data to the oracle policy by optimizing the population version. **As our second contribution**,   is to extend the PPI framework, which is originally for standard supervised setting, to the performative prediction setting. Using the example above, we can see that we need to get the true labels of the population in the optimization process to get the policy. However, sometimes, these are not known for many people without actually giving a loan. In this case, people without labels are treated as our unlabeled data. We can use LLM to get  pseudo labels for them and use the PPI framework to help promote the inference for limited gold-standard labeled data (people already known their qualifications).
>
> Lastly, as we mentioned, **we have conducted extra experiments on real data in our initial rebuttal**, hope this can help ease your concern. But one thing we want to point out is: even with real data, the reaction of population needs to be simulated, because this is an on-policy setting. That is the case for all previous performative prediction literature.

---

> > ### Comment · Reviewer_CPwq · 2025-08-09
> >
> > I thank the authors for their detailed response and will increase my score accordingly. I suggest highlighting the example earlier in the revised manuscript so that readers interested in the application can more quickly understand the setting.

---

> > > ### Author Response · Authors · 2025-08-09
> > > **Thanks for your kind response and willingness to raise the score**
> > >
> > > Dear reviewer CPwq,
> > >
> > > We appreciate your further feedback and willingness to raise the score! We really appreciate your insightful comments and suggestions. We will incorporate more detailed explanations for the motivation and highlight the concrete example.
> > >
> > > Thanks,
> > > Authors

---

### Note · Authors · 2025-08-12

We are super grateful for all the reviewers' constructive comments and willingness to raise scores. We will incorporate suggestions from the three reviewers and add extra results we demonstrated in the rebuttal, including 1. adding a motivating example early in the paper 2. finite sample analysis of variance estimation 3. a more detailed clarification on the connection to previous work and our claims on initiating the inference on performative prediction.

Thanks,
Authors

---

### Decision · Program_Chairs · 2025-09-17

**Decision:**

Accept (poster)

**Comment:**

The authors propose a new approach to statistical inference in the setting of performative prediction, where the data distribution $D(\theta)$ depends on the model parameters $\theta$, and thus one must search for a performative stable point $\theta_{PS}$ for which the parameters are optimal given its corresponding data distribution.  A typical way to obtain $\theta_{PS}$ is by repeated risk minimization (RRM): for some large number of steps ($t = 1 \ldots T$), find the loss-minimizing parameters $\hat \theta_t$ with respect to an $n$-element sample drawn from the current data distribution, then update the data distribution to $D(\hat \theta_t)$, and repeat.  The authors provide a rigorous proof of a central limit theorem for the estimate $\hat \theta_t$ (after $t$ rounds of repeated risk minimization), for large sample size $n$, showing convergence to a Gaussian with mean $\theta_t$ and variance $V_t / n$.  This is important because, combined with prior work demonstrating that $\theta_t$ converges to $\theta_{PS}$ as $t\rightarrow\infty$, it means that the estimation error in $\hat \theta_t$ does not compound and cause the RRM to diverge from the true stable point $\theta_{PS}$, i.e., repeated risk minimization will work to find $\theta_{PS}$ in practice when the typical convergence conditions are met. Moreover, the amount of estimation error can be expressed in terms of $n$ and $t$, allowing for confidence intervals or hypothesis tests for the estimate of $\theta_{PS}$. The framework is then extended to the case of prediction-powered inference (PPI), where we have a small amount of data with true labels and a larger amount of data with classifier-predicted labels, and a similar RRM procedure is conducted: the authors prove another central limit theorem result for this case.  The approach is validated with a simulation experiment of PPI under performativity in a linear regression setting.

Strengths:
* Performative prediction is an important, emerging area of machine learning.
* The work fills a critical theoretical gap, showing the effectiveness of repeated risk minimization (in practice, with finite sample sizes) for identifying the performative stable point, and providing error/uncertainty estimates that can be used for statistical inference.
* The development of the main theoretical results (explanations, theorems, and proofs) are strong, rigorous, and well-written.
* Extensions to PPI are interesting and potentially useful for real-world problems with large amounts of noisily labeled data but limited ground truth labels.

Weaknesses:
* The claim that this work "initiates the study of statistical inference under performativity" is overstated, and needs to be qualified and clarified. (The authors have agreed to tone down this claim in the final version.)
* The empirical study in the submitted version is very limited, with only a single, synthetic, and relatively simple dataset.  (The authors conducted an additional semi-synthetic study on a credit dataset during the discussion period, and will incorporate this study into the final version.)
* The motivation for extending the work from the standard performative prediction setting to performative PPI is not entirely clear, and would benefit from a real-world example where we have both large amounts of noisily labeled data and performativity.
* It took me several readings of the paper to understand why we should care about the setting with fixed $t$ and large $n$.  I now understand that this is important, both from the standpoint of what happens when $n$ and $t$ are large (successful convergence of $\hat \theta_t$ to $\theta_{PS}$) and being able to quantify the estimation error of the RRM procedure in practice (since in practice one would have to choose finite $n$ and $t$).  But it would help to provide additional clarification up front.

Reasons for accept/reject decision: This is a "borderline accept" paper with strong and important theoretical results on performative prediction, but limited empirical validation.  I lean toward acceptance because I believe that the theoretical results are sufficiently important to stand on their own (even without the very nice, additional proofs for the finite-sample case that the authors developed during the discussion period), but am willing to be overruled given some of the weaknesses above and the concerns raised by the reviewers below.   While these concerns make me hesitant to say that this is a clear accept (or that it should be given a spotlight or oral), I do think that the strengths of the paper (specifically, novelty and importance of the theoretical contributions) outweigh its weaknesses, and again, lean toward acceptance of the paper.

Discussion: The authors and reviewers had a robust and productive discussion which helped to clarify the importance of the contribution, differences from related work, and specific points in the theoretical derivation.  The authors presented both an additional, semi-synthetic case study using real-world credit data, as well as an extension of the theory to incorporate error from finite sample size $n$.  These additions would greatly benefit the accessibility and impact of the work, so it is important that they are included in the final version. After discussion, reviewers were convinced to increase scores (2 x borderline accept; 1 x accept; note that we had only three reviews for this paper).  Some minor-to-moderate concerns remained about insufficient connection to real-world examples and datasets, insufficient motivation of the performativity + PPI combination, and whether density estimation (necessary for the variance estimates $V_t$) would be effective in practice.